# Glial insulin regulates cooperative or antagonistic Golden goal/Flamingo interactions during photoreceptor axon guidance

Hiroki Takechi[1], Satoko Hakeda-Suzuki[1]*, Yohei Nitta[2,3], Yuichi Ishiwata[1], Riku Iwanaga[1], Makoto Sato[4,5], Atsushi Sugie[2,3], Takashi Suzuki[1]*

[1]Graduate School of Life Science and Technology, Tokyo Institute of Technology, Yokohama, Japan; [2]Center for Transdisciplinary Research, Niigata University, Niigata, Japan; [3]Brain Research Institute, Niigata University, Niigata, Japan; [4]Mathematical Neuroscience Unit, Institute for Frontier Science Initiative, Kanazawa University, Kanazawa, Japan; [5]Laboratory of Developmental Neurobiology, Graduate School of Medical Sciences, Kanazawa University, Kanazawa, Japan

**Abstract** Transmembrane protein Golden goal (Gogo) interacts with atypical cadherin Flamingo (Fmi) to direct R8 photoreceptor axons in the *Drosophila* visual system. However, the precise mechanisms underlying Gogo regulation during columnar- and layer-specific R8 axon targeting are unknown. Our studies demonstrated that the insulin secreted from surface and cortex glia switches the phosphorylation status of Gogo, thereby regulating its two distinct functions. Non-phosphorylated Gogo mediates the initial recognition of the glial protrusion in the center of the medulla column, whereas phosphorylated Gogo suppresses radial filopodia extension by counteracting Flamingo to maintain a one axon-to-one column ratio. Later, Gogo expression ceases during the midpupal stage, thus allowing R8 filopodia to extend vertically into the M3 layer. These results demonstrate that the long- and short-range signaling between the glia and R8 axon growth cones regulates growth cone dynamics in a stepwise manner, and thus shapes the entire organization of the visual system.

*For correspondence:
hakeda@bio.titech.ac.jp (SH-S);
suzukit@bio.titech.ac.jp (TS)

**Competing interests:** The authors declare that no competing interests exist.

## Introduction

During development, well-defined synaptic connections are formed in the brain between specific neurons to facilitate higher-order information processing. Synapses are often arranged into structures that reflect the functional organization of synaptic contacts (*Huberman et al., 2010*; *Luo and Flanagan, 2007*; *Sanes and Yamagata, 2009*). Each brain layer receives discrete axonal inputs that carry specific information. Therefore, external inputs dissolve into distinct modules in the brain. In the visual system, photoreceptors connect to columns located around the target region, thereby preserving the spatial relationships between the visual world and its representation in the brain (*Huberman et al., 2010*; *Sanes and Zipursky, 2010*). Layers separate the brain into horizontal planes, whereas columnar units group the axons into bundles that are perpendicular to the layers (*Clandinin and Zipursky, 2002*; *Mountcastle, 1997*; *Sanes and Zipursky, 2010*). The integration of the individual column and layer processes enables the modular processing of perceived information. Thus, specific layer-column axonal targeting to unique synaptic partners is a fundamental step in the complex formation of functional neuronal networks inside the brain (*Huberman et al., 2010*; *Luo and Flanagan, 2007*; *Millard and Pecot, 2018*; *Nériec and Desplan, 2016*).

The *Drosophila* visual system is an attractive model for studying the formation of the functional organization of synaptic connections because its optic ganglion has a layered and columnar structure (*Hadjieconomou et al., 2011*; *Millard and Pecot, 2018*; *Sanes and Zipursky, 2010*). The visual system of the adult *Drosophila* consists of the compound eye and four optic ganglia (in order: lamina, medulla, and lobula complex). The compound eye is composed of an array of approximately 800 ommatidia, each containing eight photoreceptor cells (R cells, R1–R8) arranged in a stereotypic pattern. R7 and R8 axons project to the second optic ganglion, namely, the medulla. The medulla is subdivided into columnar units and 10 distinct layers. R7, R8, and Mi1 axons elongate into the medulla at the earliest stage. They function as the pioneering axons during the formation of the medulla columns, which are comprised of approximately 100 different axons (*Trush et al., 2019*). R8 extends its axon to a single medulla column, followed by a single R7 axon. Eventually, R8 targets the M3 layer of the medulla, whereas R7 targets the M6 layer. Across development, the R8 neurons undergo three stages of axonal targeting (*Akin and Zipursky, 2016*; *Hadjieconomou et al., 2011*). First, single R8 axons project to a single column and form a horseshoe-shaped terminal that encircles the medulla columnar center (phase 1: third instar larva). Second, the R8 axons remain at the medulla neuropil surface without bundling with each other (phase 2: 24% APF [After Puparium Formation]). Third, R8 axons extend filopodia to target the M3 layer (phase 3: 48% APF). Many studies have detailed the molecular mechanisms that underlie the layer-specific targeting of R neurons (*Akin and Zipursky, 2016*; *Hadjieconomou et al., 2011*; *Hakeda-Suzuki and Suzuki, 2014*; *Hakeda-Suzuki et al., 2017*; *Kulkarni et al., 2016*; *Mencarelli and Pichaud, 2015*; *Millard and Pecot, 2018*; *Özel et al., 2015*). However, little is known about the formation of the medulla columnar structure.

Previous work in our lab identified a single transmembrane protein, Golden goal (Gogo), by a large-scale screen to search for genes that control R axon pathfinding (*Berger et al., 2008*). Functional studies have revealed that Gogo, with the atypical cadherin Flamingo (Fmi), guides R8 axons to the M3 layer (*Hakeda-Suzuki et al., 2011*; *Senti et al., 2003*; *Tomasi et al., 2008*). Gogo and Fmi colocalization is essential for this function. The R8 axons of *gogo* or *fmi* single mutants exhibit similar phenotypes, including defects in the axonal array due to the irregular distances between axons and the difficulty in targeting the M3 layer. Furthermore, the dephosphorylated state of a triplet Tyr-Tyr-Asp (YYD) motif in the Gogo cytoplasmic domain is important for R8 axon targeting (*Mann et al., 2012*). However, when the YYD motif is phosphorylated, Gogo appears to interfere with the ability of the R8 axon to target the M3 layer. The *Drosophila* insulin receptor (DInR), a tyrosine kinase receptor, is one of the kinases that phosphorylate the YYD motif of Gogo (*Mann et al., 2012*). A growing number of recent studies have revealed the functional involvement of DInR in nervous system development (*Fernandes et al., 2017*; *Rossi and Fernandes, 2018*; *Song et al., 2003*). Therefore, DInR may be one mechanism through which Gogo and Fmi regulate R8 axon pathfinding. Because Gogo and Fmi are conserved across *C. elegans* to humans, elucidating their role in development in *Drosophila* can greatly enhance our understanding of the molecular mechanisms of development in higher-order species.

The current study was able to examine stepwise R8 axonal targeting events across development by following protein localization and by specifically controlling Gogo and Fmi levels in R8 axons. In phase 1, Gogo and Fmi cooperated in guiding the R8 growth cone to its correct place inside the column (*gogo* function 1). In phase 2, Gogo was phosphorylated by the glial insulin signal and began to counteract Fmi to repress filopodia extension (*gogo* function 2). In phase 3, R8 axons only expressed Fmi, which directed them to the M3 layer (no *gogo* function). These results indicate that the glial insulin signal controls Gogo phosphorylation, thereby regulating growth cone dynamics, including the formation of the horseshoe shape and filopodia extension. Overall, this regulates axon-column and axon-axon interactions. Gogo possesses an interesting property wherein the phosphorylation states maintain two separate axon pathfinding functions. This is an economical strategy for increasing protein functions when there are a limited number of genes. As a result, this mechanism maintains the regular distance between R8 axons and enables the ordered R8 axonal targeting of the column.

## Results

### Gogo expression, but not Fmi expression, ceases around the midpupal stage

During development, Gogo and Fmi proteins are expressed broadly and dynamically in photoreceptors and the optic lobe. To monitor the precise expression and localization patterns of Gogo and Fmi proteins during R8 axonal targeting, knock-in flies that tag the desired proteins in a cell-specific manner with GFP or mCherry were generated using the CRISPR/Cas9 system (*Chen et al., 2014*; *Kondo and Ueda, 2013*; *Sander and Joung, 2014*). The use of these flies allowed the observation of endogenous R8 axon-specific Gogo and Fmi localization across the developmental stages between the third instar larvae and adulthood (*Figure 1*). Gogo protein was strongly expressed in the tip of R8 axons during developmental phases 1 and 2 (*Figure 1C–E*). Contrary to previous hypotheses (*Hakeda-Suzuki et al., 2011*), Gogo protein was not present during phase 3, when R8 axons filopodia elongate toward the deeper medulla layers (*Figure 1F,G* and *Figure 1—figure supplement 1*). Conversely, Fmi-mCherry expression in R8 axons was observed throughout the development stages (*Figure 1H–K*). Fmi was localized in the R8 axon tip, including thin filopodia structures during phase 3, when Gogo expression was not present (*Figure 1K*). Gogo and Fmi protein localization in the R8 axon tip during phase 1 essentially overlapped, although there were several characteristic differences (*Figure 1M–P*). Gogo-GFP signal was relatively weak in the filopodia, but accumulated at the rim of the horseshoe-shaped axon terminal that encircled the medulla columnar center (*Figure 1M', N*). On the other hand, Fmi-mCherry signal was widely distributed in the R8 axon terminal, including filopodia-like protrusions (*Figure 1M", N*). These protein localization data indicate that Gogo and Fmi functionally cooperate, so that R8 axons recognize the center of the medulla column during phase 1. The results indicate that Fmi alone promotes vertical filopodia elongation into the M3 layer during phase 3.

### Gogo and Fmi cooperatively guide R8 axons to encircle the columnar center of the medulla

R8 cell-specific strong loss-of-function (LOF) animals were generated to observe phase-specific Gogo and Fmi functions (*Figure 2*). An RNAi insertion and a heterozygous null mutation were combined (*Hakeda-Suzuki et al., 2017*), thus resulting in a strong phenotype equivalent to known *gogo* or *fmi* null mutations (*Figure 2—figure supplement 1A–F*). In the R8 cell-specific *gogo* LOF, R8 axons correctly targeted each column, but the termini intruded into the medulla columnar center and failed to form a proper horseshoe shape during phase 1 (*Figure 2A,B and D*). In phase 2, the R8 axonal termini displayed greater horizontal filopodial extension than normal, thereby enhancing the probability of encountering neighboring *gogo* loss-of-function R8 axons over time (*Figure 2B*). This excessive R8 filopodia coincides with the disrupted R8 axon termini lineup and the invasion of layers slightly deeper than M1 during phases 2 and 3 (*Figure 2E,F,H and J*). Axon bundling and incorrect targeting become more prominent later in development. As a result, multiple R8 axons (usually two) were often observed innervating a single column (yellow arrow in *Figure 2F and J*). During live imaging, vertical extension could be observed during phase 3 in tangled *gogo* loss-of-function R8 axons, thus indicating that it is difficult to uncouple axons once they have become tangled (*Figure 2—videos 1* and *2*). This can explain the observation that columnar organization becomes worse in a larger mutant area compared with a single isolated mutant axon (*Tomasi et al., 2008*).

To determine whether Gogo function in phase 2 is independent of phase 1, we performed a phase-specific knockdown of *gogo* using Gal80$^{ts}$. The temperature was changed from 18°C to 27°C during white pupal formation, so that the gogo RNAi began to be expressed after the early pupal stage. By this stage, the R8 axons that innervate the anterior half of the optic lobe had already developed a horseshoe shape as a wild type (*Figure 2—figure supplement 1H*). In phase 2, those anterior R8 axon growth cones extended longer filopodia in more radial directions than the wild type (*Figure 2O–P'''*), indicating that *gogo* loss-of-function defects observed in phase 2 were independent of those of phase 1. Altogether, these data suggest that *gogo* has two functions: column center encircling (function 1) in phase 1, and proper filopodia extension (function 2) during phase 2.

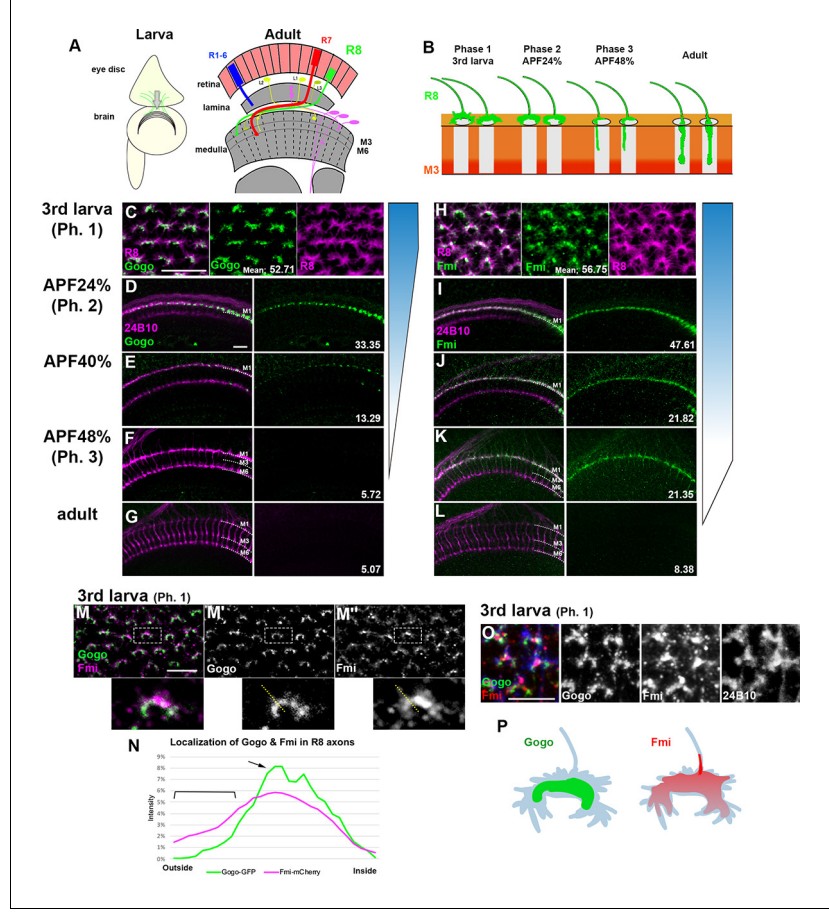

**Figure 1.** R8-specific labeling of Gogo and Fmi. (**A**) Schematics of the *Drosophila* visual system in the third instar larva and the adult. (**B**) Schematics of the phase-specific R8 targeting during development. (**C–G**) Gogo localization at the terminals of R8 axons (green) during developmental phases was visualized by combining Gogo-FsF-GFP and R8-specific FLPase (sensFLP) co-labeled with R8-specific myr-RFP (**C**) or mAb24B10 for all R axons (**D–G**) (magenta). The numbers indicate the average intensity of GFP (max. 85, *n* = 3, 24 axons each). (**H–L**) Fmi protein localization at the terminals of R8 axons (green) during developmental phases was visualized by Fmi-FsF-mCherry and R8-specific FLPase (sensFLP) co-labeled with R8-specific mCD8GFP (**H**) or mAb24B10 for all R axons (**I–L**) (magenta). The numbers indicate the average intensity of mCherry (max. 85, *n* = 3, 24 axons each). (**M–P**) Localization of Gogo (green) and Fmi (magenta) protein at the tip of the R8 axon in third instar larva (phase 1) (**M**). (**N**) The fluorescent intensity of Gogo-GFP (green) and Fmi-mCherry (magenta) was measured from outside to inside of the columns across the horseshoes as shown in M (yellow dotted lines). The average of eight axons (*n* = 3 animals) was calculated. Gogo was strongly enriched at the rim of the horseshoe-shaped R8 axon terminal (**M'**, arrow in N). Fmi was distributed broadly including filopodia (**M''**, bracket in N). 3D images of Gogo (green) and Fmi (red) localization at the tip of R8 axon (blue) in third instar larva (phase 1) (**O**). Schematic of Gogo (green) and Fmi (red) expression in R8 cells (blue) (**P**). Scale bars 10 μm.

The online version of this article includes the following source data and figure supplement(s) for figure 1:

**Source data 1.** Source data for the quantification in *Figure 1C-H*.
**Source data 2.** Source data for the quantification in *Figure 1N*.
**Figure supplement 1.** *gogo* expression gradually declines during midpupal stages.

Both of these functions were essential for avoiding axon bundling and for promoting a proper array of medulla columns during later development (*Figure 2I', J' and M*).

Similar to the *gogo* phenotype, the *fmi* LOF had R8 axon terminals that intruded into the medulla columnar center and failed to form a proper horseshoe shape during phase 1 (*Figure 2C and D*). In contrast to the *gogo* LOF, R8 filopodia horizontal extension was abnormally shortened. As a result, R8 axons maintained their distance from neighboring R8 axons and lined up orderly at the medulla

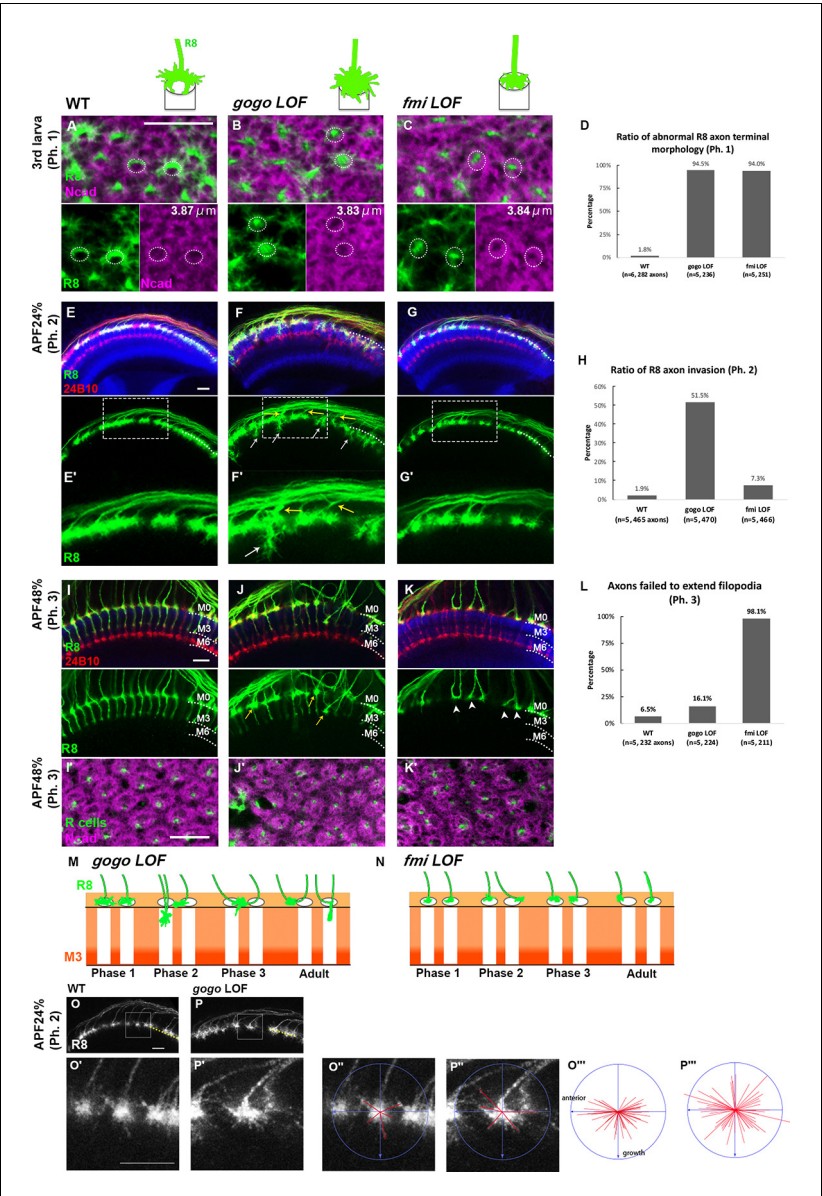

**Figure 2.** Gogo and Fmi regulates the growth cone dynamic. (**A–L**) The medulla of control, R8-specific *gogo* loss-of-function mutations, and R8-specific *fmi* loss-of-function was analyzed. (**A–C**) The medulla of the third instar larvae (phase 1) was labeled with UAS-mCD8GFP for R8 (green) and anti-N-cadherin (magenta) to visualize columns. The dashed circles demarcate columns. The numbers indicate the average diameter of the medulla columns visualized with anti-N-cadherin (*n* = 3, 18 columns). (**D**) Quantification of the R8 axon terminals that intruded into the medulla columnar center and failed to form a proper horseshoe shape during phase 1. (**E–G**) The medulla at APF24% (phase 2) was labeled with UAS-mCD8GFP for R8 (green), mAb24B10 for all R axons (red) and anti-N-cadherin (blue). *gogo* loss-of-functions showed R8 axon bundling and overextension beyond the R8 temporary layer (arrows). (**H**) Quantification of the invasion R8 axons at phase 2. (**I–K**) The medulla at APF48% (phase 3) was labeled with UAS-mCD8GFP for R8 (green), mAb24B10 for all R axons (red), and anti-N-cadherin (blue). *gogo* loss-of-function showed R8 axon bundling (arrows), whereas in *fmi* loss-of-functions, R8 axons failed to extend filopodia vertically toward the M3 layer (arrowheads). (**I'–K'**) Medulla were labeled with N-cadherin (magenta) and R axons with mAb24B10 (green) to highlight the columnar pattern. (**L**) Quantification of R8 axons that failed to vertically extend their filopodia toward the M3 layer during phase 3. (**M, N**) Schematics of R8-targeting phenotype in *gogo* loss-of-function and *fmi* loss-of-function in each phase. (**O, P**). To elucidate the function of Gogo in phase 2, *gogo* RNAi was expressed in R8 axons in *gogo* heterozygous mutant only after puparium formation (APF0%) using Gal80$^{ts}$ to eliminate the effect of *gogo* LOF in phase 1. Since the axons were sparsely labeled using Flp-out system, some axon terminals were isolated and each filopodia can be identifiable

*Figure 2 continued*

(white square in O and P. Enlarged images in O' and P'). The centers of the growth cones were plotted, and the orientation of axon growth perpendicular to boundary line of medulla was determined. Tips of the five longest filopodia were connected to the center by red lines (O'', P''). Fifty lines from ten axons were collected and merged into one image (O''', P'''). In the phase 2-specific *gogo* LOF, anterior R8 axon growth cones extended longer filopodia in more radial directions than wild type. Scale bars 10 μm.

The online version of this article includes the following video, source data, and figure supplement(s) for figure 2:

**Source data 1.** Source data for the quantification in *Figure 2A-C*.
**Source data 2.** Source data for the quantification in *Figure 2D*.
**Source data 3.** Source data for the quantification in *Figure 2H*.
**Source data 4.** Source data for the quantification in *Figure 2L*.
**Figure supplement 1.** R cell-specific loss-of-funtion of Gogo and Fmi.
**Figure 2—video 1.** Filopodial dynamics of the control animal.
https://elifesciences.org/articles/66718#fig2video1
**Figure 2—video 2.** Filopodial dynamics of *gogo* mutant.
https://elifesciences.org/articles/66718#fig2video2

surface during phase 2 (*Figure 2G and H*). Toward phase 3, R8 axons began to lose proper distance among themselves, thus resulting in defective columnar organization (*Figure 2I' and K'*). We attributed these defects to the initial failure of *fmi* R8 axons to encircle the medulla columnar center during phase 1. Moreover, in phase 3, *fmi* R8 axons failed to vertically extend their filopodia toward the M3 layer (*Figure 2K and L*). These results indicate that Gogo and Fmi function in opposing manners during phases 2 and 3 of R8 axon targeting (*Figure 2M and N*). Given that *gogo* and *fmi* LOFs had disorganized medulla columns in later stages (*Figure 2J' and K'*), it can be concluded that the column center encircling during phase 1 is important for R8 axons to follow the correct columnar path and to develop organized arrays.

## Gogo performs a cooperative and antagonistic function toward Fmi

Previous studies that are primarily based on genetic interactions have indicated that Gogo and Fmi must interact to recognize their ligand molecule (*Hakeda-Suzuki et al., 2011*). Loss-of-function mutations were used to observe any genetic Gogo/Fmi interactions during phase 1. The use of RNAi lines to knockdown each gene in an R8-specific manner resulted in morphological defects in the termini of a fraction of R8 axons (38.2% of *gogo*RNAi and 11.9% of *fmi*RNAi; *Figure 3A,B,C and E*). Double knockdown synergistically enhanced these morphological defects (76.6% of termini; *Figure 3D and E*), thus suggesting that Gogo and Fmi cooperate during phase 1 to correctly recognize and encircle the medulla columnar center.

The next set of experiments was attempted to rescue these loss-of-function mutant phenotypes by overexpressing the opposing gene to test whether Gogo and Fmi are mutually compensatory. Fmi overexpression in R8-specific *gogo* LOF did not rescue R8 axon termini morphological defects (*Figure 3—figure supplement 1I and K*). Likewise, Gogo overexpression in R8-specific *fmi* LOF did not rescue the morphological defects (*Figure 3—figure supplement 1J and L*). These results indicate that Gogo and Fmi do not have redundant gene functions and cannot compensate for each other.

To investigate the function 2 of Gogo, we examined the genetic interaction between *gogo* and *fmi* LOF in phase 2. Compared to the *gogo* single LOF, *gogo/fmi* double LOF showed much milder bundling and invasion defects in phase 2 (*Figure 3—figure supplement 1A–H*), suggesting an antagonistic function between *gogo* and *fmi* in phase 2. The antagonistic effect was more dramatic when these genes were overexpressed. When *gogo* was overexpressed in an R8-specific manner in phase 3, *gogo*-overexpressed R8 axons failed to vertically extend their filopodia toward the M3 layer, similar to that in *fmi* LOF (*Figure 3F–H*, compared with *Figure 2K*). Conversely, *fmi*-overexpressed R8 axons extended their filopodia vertically toward the layers much deeper than the wild type and passes through the medulla during phase 2 (*Figure 3I*). To observe the genetic relationship between Gogo and Fmi, Gogo levels were manipulated, and the effect on filopodia extension in *fmi*-overexpressed R8 axons was observed. *gogo* knockdown on an *fmi* overexpression background enhanced premature vertical filopodia extension during phase 2 (*Figure 3J and L*), thus resulting in

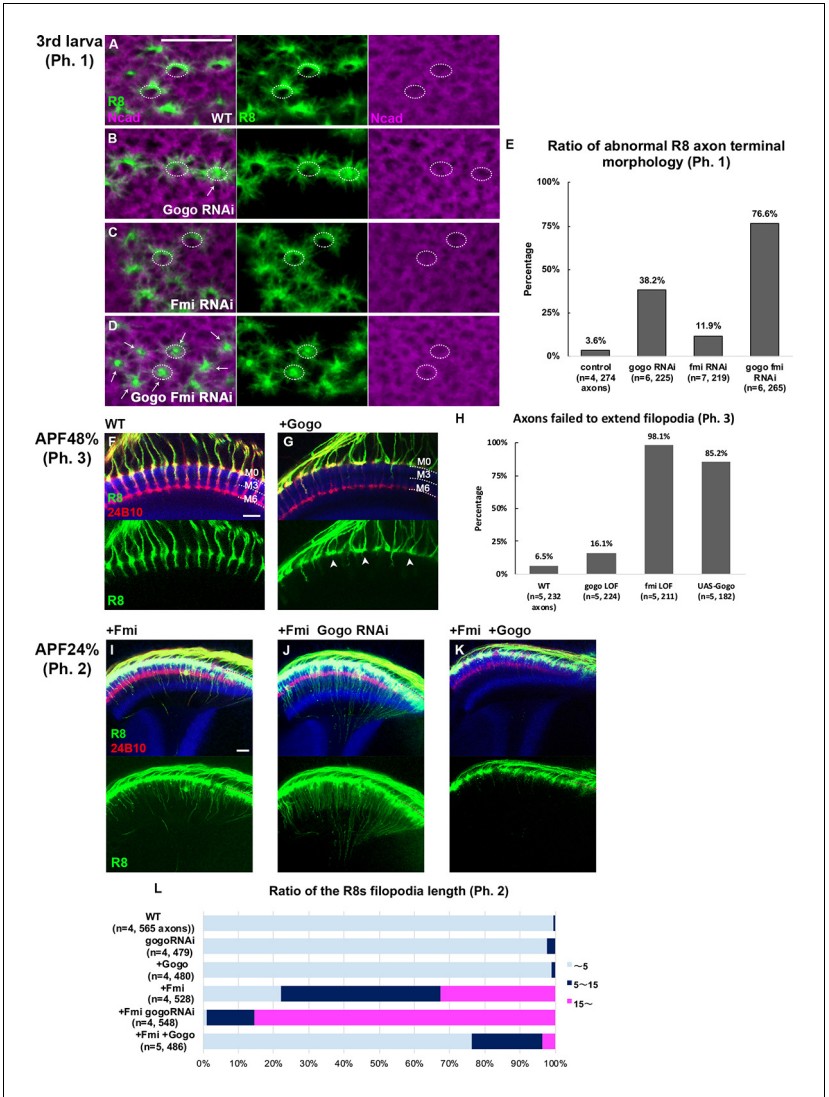

**Figure 3.** Gogo has dual functions, 'cooperative' and 'antagonistic' toward Fmi. (**A–E**) R8 axons in wild type (**A**), R8-specific knockdowns of *gogo* (**B**), *fmi* (**C**), and *gogo*, *fmi* double knockdowns (**D**) in phase 1 were visualized using R8-specific UAS-mCD8GFP (green) counterstained with anti-N-cadherin (magenta). (**E**) Quantification of the R8 axon terminals that intruded into the medulla columnar center and failed to form a proper horseshoe shape at phase 1 (third instar larva). (**F–L**) Genetic interaction between *fmi* and *gogo*. R8 axons are labeled with mCD8GFP (green), and counterstained with mAb24B10 (red) and anti-N-cadherin (blue). R8 axons overexpressing *gogo* failed to extend their filopodia vertically toward the M3 layer (arrowheads in G compared with F). (**H**) Quantification of R8 axons failed to vertically extend their filopodia toward the M3 layer during phase 3 (APF48%). (**I**) Upon *fmi* overexpression, R8 cells extended their vertical filopodia toward the deeper layer of the medulla during phase 2 (APF24%). The vertical filopodia extension was further promoted by *gogo* RNAi (**J**) and strongly suppressed by *gogo* overexpression (**K**). (**L**) Quantification of R8 filopodia length. The length of the longest filopodia was measured in 3D images and divided into three classes: <5 μm (light blue), 5–15 μm (dark blue), and >15 μm (magenta). Scale bars 10 μm.

The online version of this article includes the following source data and figure supplement(s) for figure 3:

**Source data 1.** Source data for the quantification in *Figure 3E*.
**Source data 2.** Source data for the quantification in *Figure 3H*.
**Source data 3.** Source data for the quantification in *Figure 3L*.
**Figure supplement 1.** Gogo and Fmi functions are not redundant.
**Figure supplement 2.** Functional domain analysis of Gogo.
**Figure supplement 2—source data 1.** Source data for the quantification in *Figure 3—figure supplement 2O*.

the R8 axon bundling phenotype observed at the adult stage (*Figure 3—figure supplement 1M–P*). Conversely, *gogo* and *fmi* cooverexpression suppressed filopodia extension compared with *fmi* overexpression alone (*Figure 3K and L*). These results underscore that Fmi promotes filopodia extension, which is counteracted by Gogo. Thus, as the development proceeds, Gogo genetically showed cooperative interaction (phase 1) to antagonizing interaction (phase 2) toward Fmi.

## The two functions of Gogo are regulated by the same functional ectodomain

To examine how Gogo switches its functional role regarding Fmi, we first checked if Gogo has multiple functional stretches in the extracellular domain that could elicit each function. Gogo has a GOGO domain that contains eight conserved cysteines, a Tsp1 domain, and a CUB domain in its extracellular portion. Previous work has shown that both the GOGO and Tsp1 domains are required for Gogo function (*Tomasi et al., 2008*). To determine which Gogo ectodomain is required in higher resolution, a smaller segment of each domain was deleted from the genome using CRISPR/Cas9. Severe morphological phenotypes similar to the *gogo* null mutant were observed in any of the small GOGO or Tsp1 domain deletions in phase 1 (*Figure 3—figure supplement 2A–H*). Furthermore, overexpression of the Gogo fragment lacking GOGO or Tsp1 domains showed weaker suppression of filopodia extension in the *fmi* overexpression mutants compared to the full-length Gogo overexpression (*Figure 3—figure supplement 2I–O*). These results demonstrated that GOGO and Tsp1 domains are required in both phases 1 and 2. Therefore, the same stretch of extracellular portion (GOGO–Tsp1) is required for the both functions of Gogo, indicating that switching between two functions of Gogo is not relevant to the extracellular portion during the early developmental stages.

## Gogo localization is dependent on Fmi localization inside filopodia

The functional domain in the extracellular portion of Gogo indicates that Gogo/Fmi interactions occur throughout development, including phases 1 and 2. Previous studies have shown that Gogo and Fmi colocalize at the cell-cell contacts of cultured cells (*Hakeda-Suzuki et al., 2011*). In order to test it in more in vivo situation, we tried to observe the changes of the Gogo or Fmi protein localization at phase 1 in the loss- or gain-of-function mutants (*Figure 4*). In the LOF mutants, interpretation of the localization changes was not possible because the growth cone morphology had changed drastically. Therefore, we focused on situations in which the protein was overexpressed. Fmi localization was not altered in *gogo* overexpression mutants (*Figure 4F and H*). Conversely, in *fmi* overexpression, Gogo localization shifted toward the stalk of the axon terminal, where Fmi accumulates (*Figure 4C–E*). Moreover, Gogo localization was shifted along the vertical filopodia stimulated by Fmi to prematurely extend during phase 2 (*Figure 4I and J*). These results indicate that Gogo localization is controlled by Fmi, and that the physical interaction between Gogo and Fmi controls the formation of the horseshoe structure during phase 1 and filopodia extension during phase 2.

## Dephosphorylated and phosphorylated Gogo have distinct functions toward Fmi

We next tested whether cytoplasmic domain of Gogo serves as a switch to change between its two-faced functions. Previous studies suggest that the cytoplasmic domain of Gogo is important for Gogo/Fmi collaborative functions, while they interact in *cis* (*Hakeda-Suzuki et al., 2011*; *Tomasi et al., 2008*). It has also been shown that the YYD tripeptide motif in the cytoplasmic domain is required for Gogo function (*Mann et al., 2012*). Furthermore, Tyr1019 and Tyr1020 are known as the true phosphorylation sites in vivo (*Mann et al., 2012*). To test whether regulation of Gogo phosphorylation is required for function 1 during phase 1, the Gogo phosphomimetic form (GogoDDD), non-phosphomimetic form (GogoFFD) and deletion of the entire cytoplasmic domain (GogoΔC) were used to rescue the *gogo* mutant phenotype. GogoDDD and GogoΔC were unable to rescue the mutant morphological phenotype, whereas wild-type Gogo and GogoFFD significantly rescued the phenotype during phase 1 (*Figure 5A–F*). These results indicate that the unphosphorylated YYD motif of the cytoplasmic domain is required for R8 axons to correctly recognize the medulla column and encircle the columnar center (function 1).

Next, we sought to determine which Gogo form is functional during filopodia extension in phase 2. The GogoFFD and GogoDDD transgenes were expressed in *fmi*-overexpressed flies (*Figure 5G–*

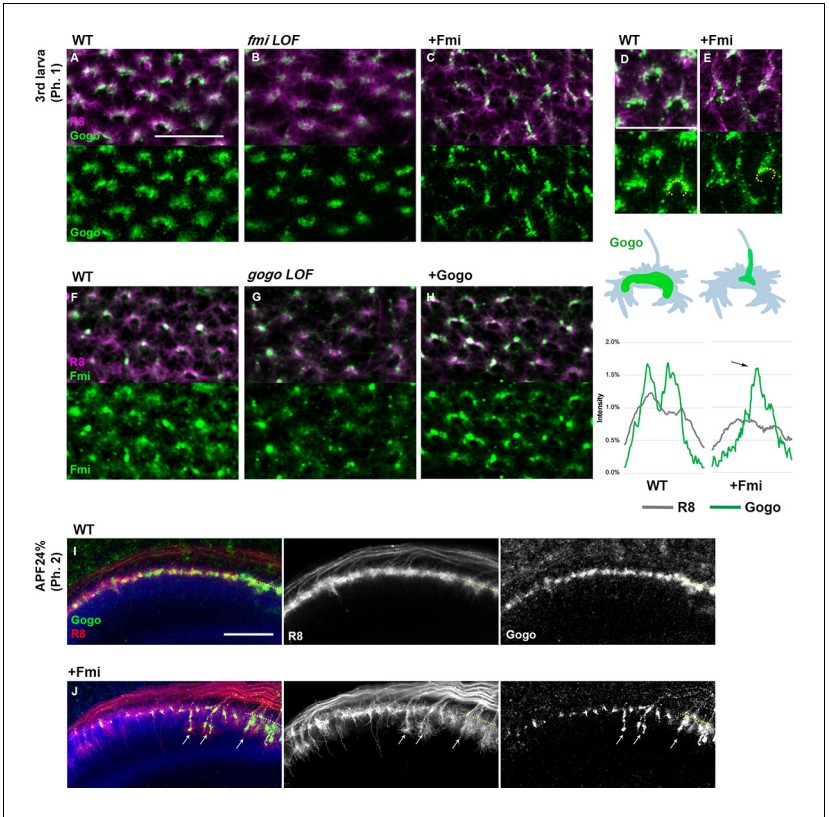

**Figure 4.** Gogo localization in R8 changes depending on the expression level of Fmi. (**A–H**) Localization of R8-specific Gogo-GFP (**A–E**) and Fmi-mCherry (**D–H**) in loss-of-function (heterozygous mutation with R8-specific RNAi) or overexpression backgrounds. R8 axons were labeled with myr-RFP or mCD8GFP. (**D–E**) 3D images of Gogo localization in R8 cells of wild type (**D**) or Fmi overexpression (**E**). The fluorescent intensity of Gogo-GFP (green) and R8 myr-RFP (gray) was measured along the horseshoe structures (the dotted lines in **D, E**) and the average of four axons (*n* = 2 animals) is shown in the graph below each image. Upon Fmi overexpression, strong Gogo expression was observed at the stalk of the axon terminal (C and E compared with A and D, arrow in the histogram of +Fmi). (**F–H**) Fmi localization did not show remarkable change in *gogo* loss-of-function (**G**) nor in *gogo* overexpression (**H**) mutants compared with the wild type (**F**). (**I, J**) R8-specific Gogo-GFP (green) during phase 2 in wild type (**I**) and Fmi overexpression mutants (**J**). R8 axons are labeled with myr-RFP (red) and counterstained with anti-N-cadherin (blue). Gogo protein was localized along the vertical filopodia that prematurely extended during phase 2 (arrows in J compared with I). Scale bars 10 μm.

The online version of this article includes the following source data for figure 4:

**Source data 1.** Source data for the quantification in *Figure 4D-E*.

---

*L*). GogoFFD did not suppress filopodia extension (*Figure 5J and L*), but GogoDDD did (*Figure 5K and L*). This indicates that the phosphorylated form of Gogo is required for filopodia suppression (function 2).

In previous studies, GogoFFD rescued the R axon targeting defects in adult stage to a considerable extent (*Mann et al., 2012*). However, in the current study at earlier stages, GogoFFD did not completely rescue ectopic filopodia extension and axon bundling, thus resulting in a slightly premature R8 termini intrusion into the medulla neuropil during phase 2 (*Figure 5M–Q*). Therefore, Gogo phosphorylation may occur sometime between phases 1 and 2 to suppress excessive filopodia formation and extension during normal R8 axon development. These results suggest that non-phosphorylated Gogo governs function 1, while the phosphorylated form controls function 2, and that each phosphorylation state has a decisive function in axon pathfinding to form complex functional neuronal circuits.

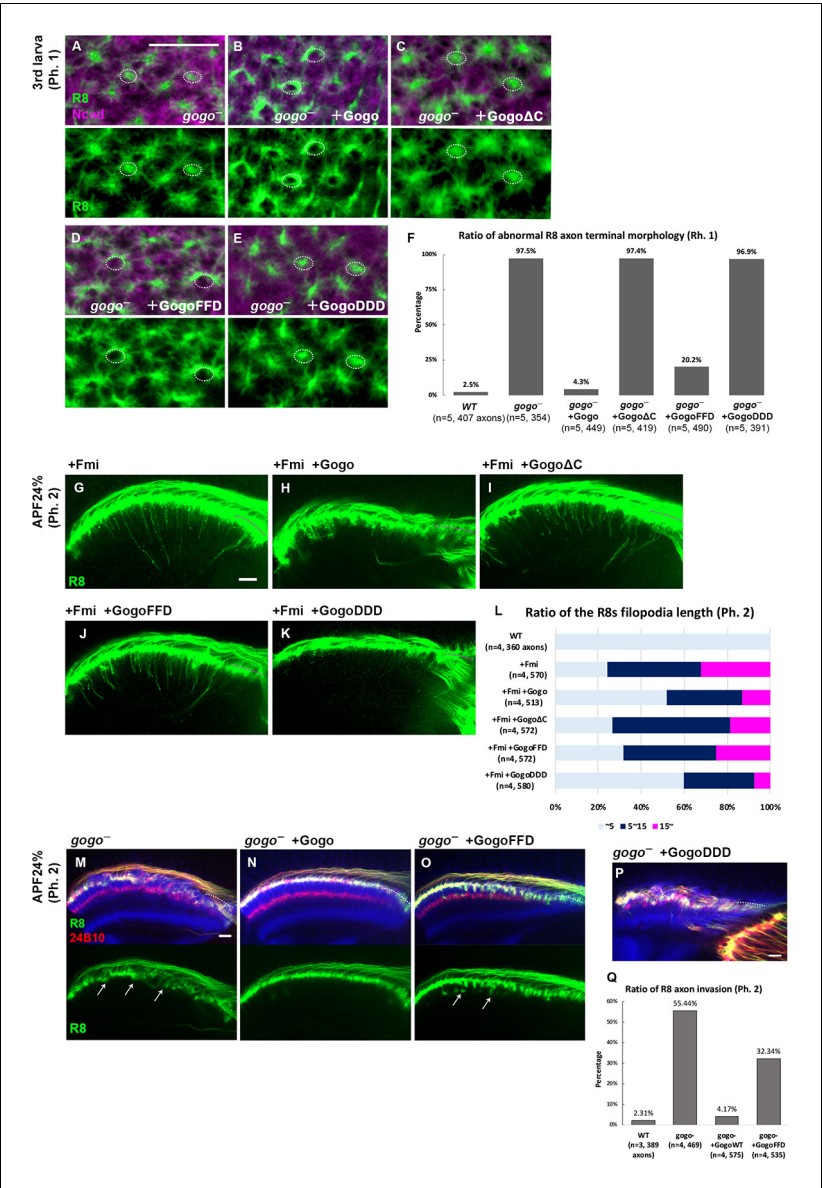

**Figure 5.** Dual function of Gogo controlled by the phosphorylation of YYD motif. (**A–F**) *gogo* rescue experiments in a background of *gogo*[H1675]/*gogo*[D1600] during phase 1 (third instar larva). R8 axons were visualized with mCD8GFP (green), and columns were labeled with N-cadherin (magenta). The targeting defects of *gogo* mutants (**A**) were almost completely rescued by wild-type Gogo (**B**) and GogoFFD (D, non-phosphomimetic), but not rescued by GogoΔC (**C**) or GogoDDD (E, phosphomimetic). (**F**) Quantification of R8 axon terminals that intruded into the medulla columnar center and failed to form a proper horseshoe shape at phase 1 (third instar larva). (**G–L**) Horizontal images of R8 axons expressing GogoFFD or GogoDDD in an Fmi overexpression background at phase 2 (APF24%). R8 filopodia elongation was significantly repressed by wild-type Gogo (**H**) or GogoDDD (**K**), but not by GogoΔC (**I**) nor GogoFFD expression (**J**). Quantification of R8 axon filopodia length (**L**). The length of the longest filopodia in a 3D image was measured and divided into three classes: <5 μm (light blue), 5–15 μm (dark blue), >15 μm (magenta). (**M–Q**) Ectopic filopodia extension and axon bundling (arrows in M) in *gogo* mutants (*gogo*[H1675]/*gogo*[D1600]) were rescued by wild-type Gogo (**N**), but not by GogoFFD expression (arrows in O) during phase 2 (APF24%). (**P**) The R8 axons in GogoDDD-rescued animals were too disrupted to be quantified. (**Q**) Quantification of the R8 axon invasion during phase 2. Scale bars 10 μm.

The online version of this article includes the following source data and figure supplement(s) for figure 5:

**Source data 1.** Source data for the quantification in *Figure 5F*.
**Source data 2.** Source data for the quantification in *Figure 5L*.
**Source data 3.** Source data for the quantification in *Figure 5Q*.

*Figure 5 continued on next page*

*Figure 5 continued*
**Figure supplement 1.** Gogo and Fmi cytoplasmic domain change its functional properties.
**Figure supplement 1—source data 1.** Source data for the quantification in *Figure 5—figure supplement 1B*.
**Figure supplement 1—source data 2.** Source data for the quantification in *Figure 5—figure supplement 1E*.
**Figure supplement 1—source data 3.** Source data for the quantification in *Figure 5—figure supplement 1H*.

## Suppression of Fmi by phosphorylated Gogo is mediated via adducin

Gogo interacts with the actin-capping protein Hu-li tai shao (Hts, *Drosophila* adducin homolog) to control R8 neuron axonal extension (*Ohler et al., 2011*). Thus, we hypothesized that function 2 of Gogo, which suppresses filopodia, relies on the actin-capping ability of Hts. Thereafter, R8-specific *hts* LOF was analyzed. During phase 2, *hts* LOF R8 axon termini had excessive radial filopodia extensions and an axon-axon bundling phenotype similar to *gogo*[-/-] mutants (*Figure 5—figure supplement 1A and B*), suggesting that Hts works with Gogo to prevent excessive filopodia extension. To determine which Gogo form works with Hts, Hts was co-overexpressed with GogoDDD or GogoFFD, and observed during phase 3 (*Figure 5—figure supplement 1C*) and in adulthood (*Figure 5—figure supplement 1D*). Wild-type Gogo or GogoDDD overexpression partially suppressed filopodia extension (*Figure 5—figure supplement 1C*). GogoDDD/Hts coexpression, but not GogoFFD/Hts coexpression, synergistically suppressed filopodia extension or resulted in R8 axon stalling at the medulla surface layers (*Figure 5—figure supplement 1C–E*). These data indicate that phosphorylated Gogo sends signals via Hts to suppress filopodia extension.

## Glial cell insulin signal is critical for Gogo phosphorylation

The Gogo/Fmi interaction phenotype can be considered 'cooperative' in function 1 (phase 1) but changes to 'antagonistic' in function 2 (phase 2) (*Figures 2* and *3*). This indicates that Gogo is phosphorylated during the transition from functions 1 to 2, but the mechanism is unclear. Previous work indicates that DInR phosphorylates Gogo and is important for its function (*Mann et al., 2012*). DInR has tyrosine kinase activity and is known to phosphorylate the YYD motif. Therefore, R8-specific *dinr* LOF was created. The *dinr* LOF did not have defects in phase 1 (*Figure 6A*). During phase 2, the *dinr* LOF R8 axons displayed a similar phenotype to the GogoFFD rescue and exhibited radial filopodia extensions, thus resulting in R8 axon bundling and the premature invasion of the deeper medulla layers (*Figure 6A–C*, compare with *Figure 5D and O*).

We next sought to determine how DInRs on R8 axons receive insulin signals. Previous gene expression studies in the developing optic lobe revealed that among the eight *dilp* genes, *dilp6* is expressed in glial cells in *Drosophila* (*Fernandes et al., 2017*; *Okamoto and Nishimura, 2015*; *Rossi and Fernandes, 2018*; *Sousa-Nunes et al., 2011*). By using Gal4 lines, *dilp6* was confirmed to be expressed in the surface and cortex glia at all developmental stages (*Figure 6D* and *Figure 6—figure supplement 1A–I*). To identify whether glia contributes to Gogo phosphorylation in R8 axons, glial-specific protein secretion was blocked during phase 2. Dynamin is known to control peptide secretion, including insulin-like peptides (*Wong et al., 2015*). The temperature-sensitive dynamin mutant (*shibire*[ts1] [*shi*[ts1]]) was specifically expressed in glial cells to block Dilp secretion. This produced a defective phenotype similar to the *dinr* LOF; R8 axons showed radical filopodia extensions and bundling with premature invasion into deeper medulla layers (*Figure 6E and F*). These defects were also observed when *shi*[ts1] was specifically overexpressed in surface and cortex glial cells (*Figure 6G,J and M*). Conversely, we could not see any defects when we block the protein secretion from insulin-producing cells (IPC) (*Figure 6K and M*) or other types of glial cells, including medulla neuropil glia and Chiasm glia (*Figure 6H,I and M*).

The *hobbit* gene is known to regulate Dilp secretion (*Neuman and Bashirullah, 2018*). Therefore, *hobbit* was knocked down to block Dilp secretion specifically in glial cells. This produced a similar phenotype as the *dinr* LOF, thus supporting the idea that glial Dilp controls R8 axonal targeting (*Figure 6L*).

We further investigated the genetic interaction between *dilp6* and Fmi overexpression (*Figure 6N–Q*). Fmi overexpression counteracted phosphorylated Gogo, and created the sensitized background to study Gogo function 2 (*Figure 5K*). In this background, we found that *dilp6* RNAi

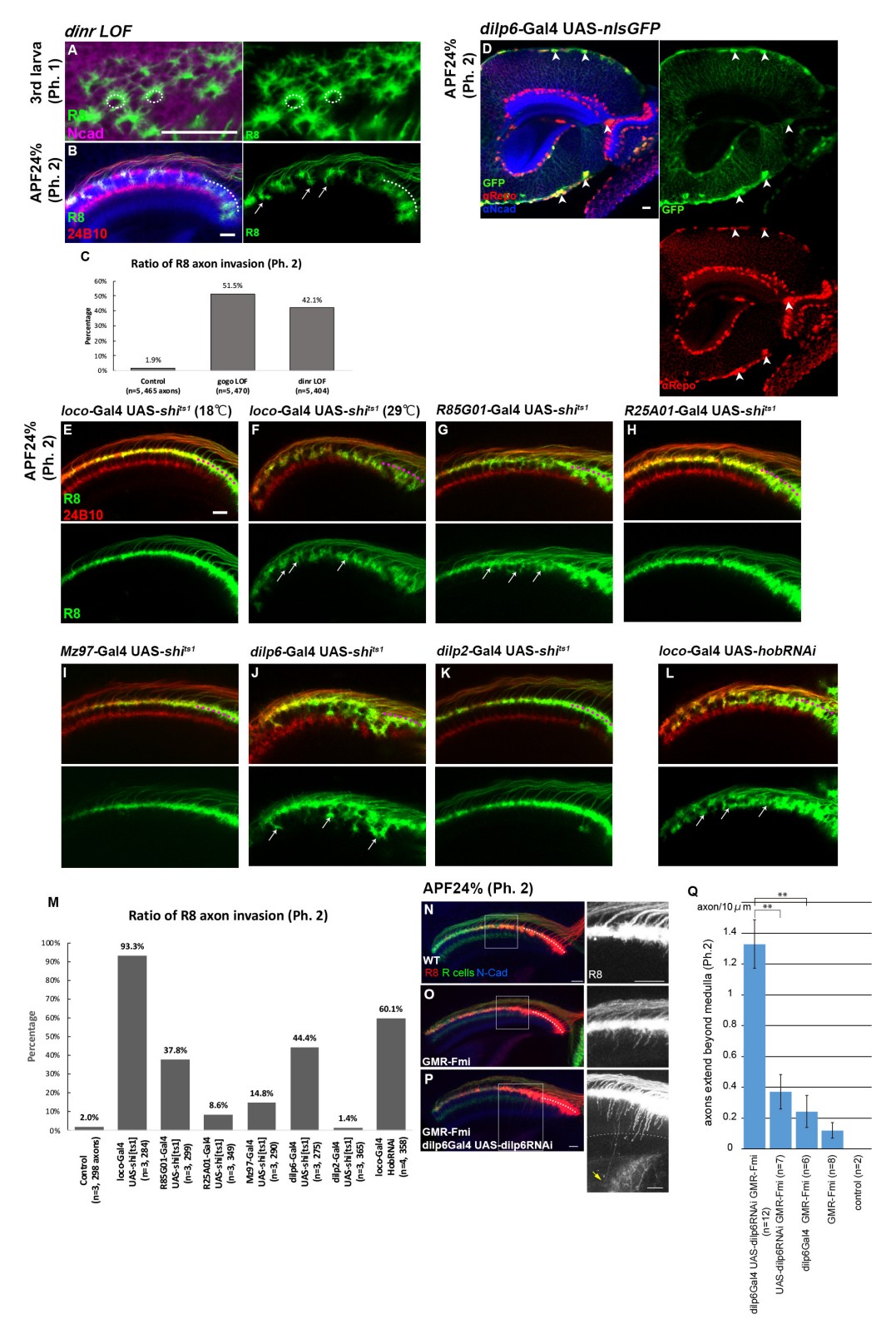

**Figure 6.** Glial insulin switches the Gogo-Fmi function from 'cooperative' to 'antagonistic'. (A–C) The phenotype of R8-specific *dinr* loss-of-function (*dinr* heterozygotes with R8 cell-specific RNAi) at the third instar larvae and APF24% (phase 1 and 2) was analyzed using R8-specific mCD8GFP (green) counterstained with mAb24B10 (red in B) and anti-N-cadherin (magenta in A, blue in B). R8 axons bundled together, resulting in invasion into deeper medullar layers in phase 2 (arrows in B). (C) Quantification of the R8 axon invasion during phase 2. (D) Dilp6-Gal4 expression monitored by nuclear GFP

*Figure 6 continued on next page*

*Figure 6 continued*

reporter (green) was mainly observed in cortex and surface glial cells in the optic lobe during phase 2 (arrowheads). Glial cells were labeled with anti-repo (red), and optic neuropils with anti-N-cadherin (blue). (E–K) The secretion of the Dilp was blocked in cells expressing UAS-*shi*^ts1 using *loco*-Gal4 (E and F) in all glial cells, GMR85G01-Gal4 (G) in surface and cortex glia, GMR25A01-Gal4 (H), Mz97-Gal4 (I) in wrapping and neuropil glia, *dilp6*-Gal4 (J) and *dilp2*-Gal4 (K). During phase 2, R8 axons labeled by myr-tdTomato (green) showed the bundling phenotype in surface and cortex glia-specific *shi*^ts1 expression (arrows in F, G, and J). Although these Gal4 drivers were expressed from the larval stages, the effect of blocking by shi[ts] began from APF0% when the temperature was shifted to 29°C. (L) Glia-specific inhibition of Dilp secretion by *hobbit* RNAi expressed under a *loco*-Gal4 driver. R8 axons bundled with each other, resulting in invasion into the deeper medullar layers (arrows). (M) Quantification of R8 axon invasion in E–L. (N–Q) To investigate the genetic interaction between glial *dilp6* and filopodia extension during phase 2, *dilp6* RNAi was expressed in glial cells using *dilp6*-Gal4, and Fmi was overexpressed in photoreceptors using GMR-Fmi. R8 axons were visualized using myr-tdTomato (red, white in the right side of each panel) together with all photoreceptor axons (green) and N-cadherin (blue). GMR-Fmi flies showed enhanced filopodia extension (O). Knockdown of *dilp6* using *dilp6*-Gal4 and UAS-*dilp6*RNAi significantly enhanced the phenotype (P), and several filopodia extended over the medulla (arrow). The dotted line indicates the lower edge of the medulla. (Q) Quantification of the number of axons that extend over the medulla. Medulla region was determined according to the Ncad staining. Total number of the filopodia extensions beyond the medulla were counted from several images, and the average number per 10 μm section was calculated. **p<0.001, Welch's t-test. Scale bars 10 μm.

The online version of this article includes the following source data and figure supplement(s) for figure 6:

**Source data 1.** Source data for the quantification in *Figure 6C*.
**Source data 2.** Source data for the quantification in *Figure 6M*.
**Source data 3.** Source data for the quantification in *Figure 6Q*.
**Figure supplement 1.** Related to Figure 6 *dilp* genes expression pattern in optic lobe.

knockdown combined with *dilp6*-Gal4 expression (driver for surface and cortex glia) could further enhance the defects caused by Fmi overexpression (*Figure 6P and Q*).

Taken together, these results suggest that glial Dilp6 at least partially mediates the Gogo phosphorylation signal into R8 axons. Thus, taken together, the data indicate that in R8 neurons, DInR phosphorylates the Gogo cytoplasmic YYD motif upon receiving glia-derived insulin signals during phase 2.

## Glia supplies Fmi that interacts with R8 axons in the columnar center

We have shown that Gogo and Fmi direct R8 axons to recognize the columnar center. However, the component that R8 recognizes during phase 1 is unclear. We hypothesized that the Fmi located on R8 axons functions as a cadherin and homophilically adheres with Fmi on neighboring cells, thereby allowing R8 axons to correctly target the medulla. R7, R8, and Mi1 neurons are known to be the core members during the earliest medulla column formation step (*Trush et al., 2019*). To test whether functional Fmi is located on R7 or Mi1, Fmi was specifically knocked down in R7 or Mi1 neurons. This did not result in detectable defects in the overall R8 axon targeting or termini morphology (*Figure 7—figure supplement 1A and B*). During the analysis of glial cell function for insulin signaling, we noticed a firm localization of the Fmi protein at the glial protrusion in the columnar center at phase 1 (*Figure 7A and B*).

The glial protrusion seemed to extend into the medulla layers as early as the entry of the R8 growth cone (arrowhead in *Figure 7A*). The protrusion passes the R8 growth cone and extends deeply into the medulla layers. However, it starts to retract towards the late third instar of larvae (yellow arrow in *Figure 7A*), and completely retracts from medulla layers in APF24% (phase 2) (*Figure 7C′*).

Considering that glial cells also contact R8 axons, glia-specific *fmi* LOF were created. Strikingly, the phenotypes were similar to that of the *gogo* and *fmi* R8 LOFs (*Figure 2D and H*). R8 axon termini in the optic lobe of these mutants failed to encircle the columnar center and intrude into the central area (*Figure 7C–E*), but no bundling at phase 2 (*Figure 7C′ and D′*). In the phase 3, columnar organization was disturbed as well. Proper distance was not maintained between R8 axons and the fine columnar array was disrupted in glia-specific *fmi* LOF (*Figure 7C″ and D″*).

Changes in R8 axon Gogo and Fmi localization were analyzed in glia-specific *fmi* knockdowns to further assess the functional relationship between glial Fmi and R8 Gogo/Fmi. In this knockdowns, R8 axon Fmi localization was weaker in the filopodia tips and accumulated in the axon termini stalk (*Figure 7F and G*). R8 axon Gogo localization was more diffuse throughout the entire termini structure, including the filopodia (*Figure 7H and I*). These localization changes indicate that Gogo and

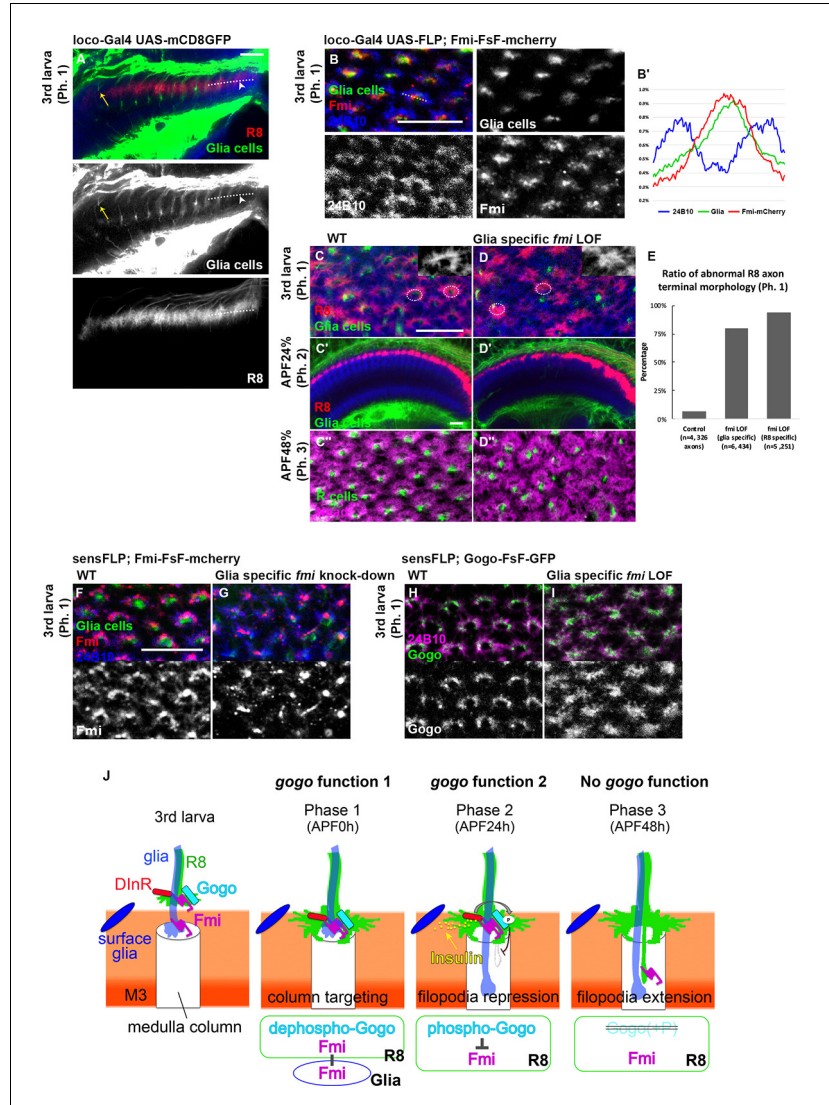

**Figure 7.** Glial Fmi and R8 Gogo/Fmi instruct R8 to recognize the columnar center. (**A**) R8 axon terminals visualized with myr-tdTomato (red, white) and glial cells visualized with mCD8GFP (green) and counterstained with anti-N-cadherin (blue) in phase 1 (third instar larva). The glial protrusion extended into the medulla layers as early as the R8 growth cone enters (arrowhead). In the oldest column, the glial protrusions have begun to retract (yellow arrow). (**B**) Fmi protein localization at the terminals of glial cells (red) was visualized by Fmi-FsF-mCherry and glial-specific FLPase (loco-Gal4 UAS-FLP) co-labeled with glial-specific mCD8GFP (green) and mAb24B10 for all R axons (blue) in phase 1 (third instar larva). The fluorescence intensity of Fmi-mCherry (red), glial-specific mCD8GFP (green), and stained R axons (blue) was measured across the column (dotted lines) and the average of eight axons (*n* = 3 animals) was shown in the graph (**B'**). (**C–E**) Medulla of the wild type (**C**) and glial-specific *fmi* loss-of-function (*fmi* heterozygote with glial cell-specific RNAi [*loco*-Gal4, UAS-RNAi, at 29℃]) (**D**) at each phase (third instar larvae, APF24%, 48%). Labeling is the same as in (**A**). The medulla columnar pattern is labeled with N-cadherin (magenta) and R axons with mAb24B10 (green). In glial-specific *fmi* loss-of-function, R8 axon terminals intruded into the medulla columnar center and failed to form a proper horseshoe shape during phase 1 (**D**), but no bundling was observed during phase 2 (**D'**). The columnar array was disrupted at APF48% (phase 3) (**D''**). (**E**) Quantification of the R8 axon terminals that intruded into the medulla columnar center and failed to form a proper horseshoe shape at phase 1 (third instar larva). (**F, G**) The protrusions of glial cells (green) in medulla neuropils and Fmi-mCherry (red) in R8 cells were visualized in phase 1 (third instar larva). R axons were labeled with mAb24B10 (blue). (**H, I**) Localization of R8 specific Gogo-GFP (green) in glia-specific *fmi* loss-of-function. R axons are labeled with mAb24B10 (magenta) in phase 1 (third instar larva). (**J**) Model for the interaction between dual-function Gogo and Fmi to navigate R8 axons. In phase 1, non-phosphorylated Gogo/Fmi at R8 termini interact in trans with Fmi localized on the glial surface to correctly recognize the medulla columnar center (*gogo* function 1). In phase 2,

*Figure 7 continued on next page*

*Figure 7 continued*
Gogo is phosphorylated dependent on insulin signaling derived from surface and cortex glia. Phospho-Gogo antagonizes Fmi, thereby suppressing filopodia extension (*gogo* function 2). In phase 3, Fmi alone brings the R8 axon to the M3 layer, since Gogo protein is no longer expressed in R8 axons by this phase (no *gogo* function). Scale bars 10 µm.

The online version of this article includes the following source data and figure supplement(s) for figure 7:

**Source data 1.** Source data for the quantification in *Figure 7E*.
**Figure supplement 1.** Gogo and Fmi interact with Fmi to regulate cytoskeletal reorganization.
**Figure supplement 1—source data 1.** Source data for the quantification in *Figure 7—figure supplement 1J*.
**Figure supplement 2.** Genetic interaction between *gogo* and *fmi* in mushroom body.

Fmi relocate from the R8 axon horseshoe rim to other regions when R8 axon Fmi cannot bind to glial Fmi. These results also indicate that the in trans interaction between glial Fmi and R8 Gogo/Fmi mediates precise R8 axon recognition of the medulla columnar center, including the formation of a horseshoe structure. Therefore, the phenotypes described here may be the consequence of the specific interruption of function 1, but not function 2 of Gogo. In other words, this glial Fmi and R8 Gogo interaction is mediated by non-phosphorylated Gogo, and later the phosphorylation of Gogo switches the Gogo/Fmi function from 'collaborative' (function 1) to 'antagonistic' (function 2) (*Figure 5*).

Taken together, these results suggest that the glial insulin signal controls the phosphorylation status of Gogo, which regulates the growth cone dynamics of R8 and mediates axon-glia and axon-axon interactions (*Figure 6*). This mechanism maintains a consistent distance between R8 axons, enables ordered R8 targeting of the column, and eventually contributes to the formation of the organized array of the medulla columns (*Figure 7J*).

## Discussion

The current study demonstrated that R8 axons are guided in a stepwise manner via Gogo/Fmi interactions that initially have a collaborative relationship, which later becomes antagonistic during the development of the visual system (*Figure 7J*). During phase 1, dephosphorylated Gogo interacts with Fmi in *cis*, and cooperatively functions to navigate R8 axons to the correct target. During this stage, R8 Gogo interacts with glial Fmi to locate the column center and enable R8 axon terminals to form a horseshoe-like morphology that encircles the central area of the medulla column. During phase 2, Gogo is phosphorylated by the insulin signal derived from the surface and cortex glia. Phosphorylated Gogo antagonizes Fmi via Hts (adducin) to suppress filopodia extension. During phase 3, Gogo is no longer expressed in R8 axons; therefore, Fmi alone navigates R8 axons to the M3 layer. Two Gogo states control axon-axon interaction to maintain R8 axon distance and axon-column interaction for proper column targeting.

Similar Gogo/Fmi interactions are broadly utilized in the *Drosophila* nervous system. Previous work has shown that Gogo and Fmi function in dendrite formation during the embryonic stage (*Hakeda-Suzuki et al., 2011*; *Hakeda and Suzuki, 2013*). Additionally, phenotypic and genetic interaction analysis of *gogo*/*fmi* mutants/knockdowns in the mushroom body (MB) revealed that Gogo and Fmi functionally cooperate or antagonize depending on the context to regulate correct axon targeting similar to visual system (*Figure 7—figure supplement 2*). The MB is a higher center for olfactory learning and memory (*de Belle and Heisenberg, 1994*). Previous studies have shown that *fmi* mutant axons also have targeting defects in MB neurons (*Reuter et al., 2003*). Given that Fmi is broadly functionally conserved among species (*Berger-Müller and Suzuki, 2011*; *Rapti et al., 2017*; *Shi et al., 2014*; *Tissir et al., 2002*), elucidating the conserved function of Gogo/Fmi interactions in the *Drosophila* brain can provide valuable insights into the formation of higher-order nervous systems, such as the mammalian brain.

### Gogo and Fmi cooperatively mediate R8 axon-column interaction in function 1 (phase 1)

During phase 1, R8 axon terminals form a horseshoe-like shape and encircle the medulla column center. In this phase, Gogo and Fmi protein localize at the R8 axon terminal fringe surrounding the

medulla center and appear to interact in *cis* (*Figure 1M*). Because GogoFFD rescued the *gogo* mutant phenotype at this time point, it can be deduced that only the non-phosphorylated version is required (*Figure 5D–F*).

We asked what does phosphorylation do to the function of Gogo. Gogo/Fmi interactions in *cis* occur with the same affinity regardless of the Gogo phosphomimetic version in S2 cultured cells (*Mann et al., 2012*). Furthermore, GogoDDD and GogoFFD localization did not differ in the R8 axon termini during phase 1 in vivo (*Figure 5—figure supplement 1F*), suggesting that the phosphorylation status of Gogo does not change the molecular affinity of Gogo/Fmi.

Gogo phosphorylation may control multiple aspects of this process, including downstream Gogo/Fmi intracellular signaling. The Fmi downstream signaling pathway components that regulate dendrite formation or planar cell polarity (PCP) are well known (*Berger-Müller and Suzuki, 2011*; *Kimura et al., 2006*; *Li et al., 2016*; *Lu et al., 1999*; *Usui et al., 1999*; *Wang et al., 2016*). Previous studies have shown that PCP complex mutants display normal R8 axon targeting in adulthood (*Hakeda-Suzuki et al., 2011*). Moreover, the RNAi knockdown of components that are thought to regulate the dendrite formation downstream of Fmi, such as PCP complexes and G alpha proteins, did not result in defective R8 axon targeting phenotypes (data not shown). Functionally, the deletion of the intracellular domain of Fmi can promote filopodia elongation but does not mediate column center encircling (*Figure 5—figure supplement 1G–I*). Given that the Gogo cytoplasmic domain is also required for column center encircling (*Figure 5C*), the Gogo/Fmi interaction in phase 1 may send signals via both Gogo and Fmi cytoplasmic domains.

Previous studies have reported that Gogo/Fmi co-overexpression in R7 axons redirects them to the M3 layer. This occurs when GogoFFD, but not GogoDDD, is expressed (*Mann et al., 2012*). The observation of this redirection process showed that R7 axons do not extend in a stepwise manner such as R8 axons but retreat to the M3 layer from M6 (*Figure 7—figure supplement 1C and D*). This indicates that Gogo/Fmi co-overexpression does not form a code for M3 targeting but promotes cytoskeletal reorganization, which might lead to R7 axon retraction. Consistent with this idea, R7 retraction was recapitulated by overexpressing Rho by using GMR-Rho1 (*Figure 7—figure supplement 1F*). It is well known that Rho promotes cytoskeletal reorganization by activating caspase (*Aznar and Lacal, 2001*; *Barrett et al., 1997*; *Mashima et al., 1999*; *Shi and Wei, 2007*; *Sokolowski et al., 2014*). The retraction ratio was also enhanced by co-overexpressing Gogo (*Figure 7—figure supplement 1H and J*).

Strong Gogo/Fmi co-overexpression results in serious cell death in the retina (*Tomasi et al., 2008*), with greater cell death in GogoFFD than in GogoDDD. If these cell deaths are the result of increased cytoskeleton reorganization, it may indicate that GogoFFD and Fmi cooperatively regulate the cytoskeleton ectopically in various phases throughout photoreceptor development. This cytoskeletal reorganization mediated by GogoFFD might regulate the cytoskeleton in a similar manner when R8 axon Gogo/Fmi interact with glial Fmi to form the horseshoe structure during phase 1 (*Figures 2* and *7*). However, the manner in which GogoFFD sends signals via downstream components and regulates cytoskeleton reorganization is unknown; this must be addressed in the future.

## Glia interact with R8 cells to guide R8 axons in function 1 (phase 1)

This study shows that Gogo/Fmi at the R8 termini interacts in trans with Fmi, which is localized on the glial surface during phase 1 (*Figure 7*). Related to these findings, N-cadherin (Ncad) plays a role in medulla column formation (*Trush et al., 2019*). *Ncad* mutant R8 axons have a defect in targeting the medulla column, which is thought to be due to the difference in adhesive properties of the axons in the column, that is, the differential adhesion hypothesis (DAH) (*Foty and Steinberg, 2005*; *Murakawa and Togashi, 2015*; *Trush et al., 2019*). In this system, axons with greater Ncad expression tend to target the center of the column, whereas those with lower expression tend to surround the edge of the column border. Ncad overexpression in R8 axons results in changes in termini morphology and in the coverage of the entire medulla column surface (*Trush et al., 2019*).

In the current studies, Fmi overexpression in the R8 axon termini did not change the horseshoe shape (*Figure 3—figure supplement 1I*). However, *fmi* LOF in R8 axons resulted in misguided filopodia invading the column center; this does not support the DAH theory for Fmi (*Figure 2C*). Therefore, we suggest that as a cadherin, Fmi interacts homophilically in trans as Fmi/Fmi between glia and R8 cells. Conversely, Gogo interacts with Fmi in *cis* to form Gogo/Fmi on the R8 membrane.

Distinct signaling regulation via Gogo and Fmi cytoplasmic domains enables R8 axons to correctly target the medulla column.

One interesting observation is that Gogo localization differed between R8 axon- and glial-specific *fmi* LOFs: Gogo protein localization is more diffuse in R8 *fmi* LOF than in glial *fmi* LOF (*Figures 4B* and *7I*). It is known that Gogo and Fmi do not interact in trans, which was shown in cell culture systems (*Hakeda-Suzuki et al., 2011*). These observations suggest that Gogo/Fmi is not only interacting with glial Fmi, but the Gogo ligand (factor X) exists on the glial membrane and interacts with Gogo as Gogo/factor X, in addition to the Fmi/Fmi interaction. The functional role of factor X on glial cells is unknown. Therefore, it is important to identify the role of factor X to reveal the functional significance of glial-derived signaling during phase 1 of R8 axon targeting.

## Temporal and spatial regulation of Gogo phosphorylation status by glia

In phase 1, R8 axons interact with Fmi on glial cells. In phase 2, R8 axons receive insulin from surface and cortex glia. However, insulin expression started at the transcriptional level during phase 1 (*Figure 6—figure supplement 1H and I*); therefore, the temporal relationship of Gogo phosphorylation and insulin expression onset does not match apparently.

One explanation is that it is regulated via changes in the relative position between the glia and medulla during development. Glial position changes across phases 1 to 2 as the entire brain structure changes. There is a huge distance between glia and the medulla neuropil during phase 1 that drastically shrinks by phase 2. This physical distance between glia and R8 axon termini might influence the reception efficiency of insulin.

The second explanation is that there might be a slow transition between the non-phosphorylated state to the phosphorylated state. Gogo coexists as two phosphorylated states in the tip of R8 axons when R8 axons reach the medulla column. Only the microlocalization of the two phosphorylated states might be differently regulated. The shape of the growth cone was shown to be different between GogoFFD rescue and wild type rescue in the *gogo* mutant during phase 1 (*Figure 5B and D*). This difference might be due to Gogo phosphorylation and may occur even in wild type overexpression that gained the ability to suppress filopodia extension in phase 1.

The transition of total Gogo protein levels in the R8 axons also appeared to be slow. This is based on the observation that *gogo*-Gal4 strain, in which Gal4 is knocked into the *gogo* intron locus by using the MiMIC system (*Venken et al., 2011*), loses GFP protein levels (monitored by UAS-mCD8GFP, *Figure 1—figure supplement 1*) gradually, similar to the gradual decrease of Gogo-GFP fusion protein during the midpupal stages. This indicates that Gogo protein perdurance is similar to mCD8GFP and is not actively degraded by the ubiquitin-proteasome pathway. In summary, in contrast to the stepwise regulation of R8 axon extension that occurs in precise temporal phases, the slow transition of Gogo phosphorylation and the protein level decrease seem not to be the only regulatory signals that determine whether R8 axons are extended or not.

## Gogo acts antagonistically against Fmi in R8 axon-axon interactions in function 2 (phase 2)

Filopodia are formed by actin polymerization. If the concentration is above a specific threshold, in vitro experiments suggest that actin can polymerize itself. The actin concentration in vivo is typically higher than the threshold. This suggests that actin should primarily be controlled by factors that interfere with or suppress uncoordinated actin fiber polymerization in the R8 axon growth cone (*Pantaloni et al., 2001*; *Pollard and Borisy, 2003*). To prevent filopodia extension, actin-capping proteins bind to the end of F-actin, which blocks further actin fiber polymerization. The current study showed that phosphorylated Gogo activates the actin-capping protein Hts to prevent uncontrolled actin polymerization in R8 axon termini (*Figure 5—figure supplement 1C–E*). The overexpression of Hts in R8 axons alone did not prevent R8 filopodia extension, thus suggesting that phosphorylated Gogo is required. However, a previous cell culture study demonstrated that physical Gogo/Hts interactions take place regardless of the phosphorylation status of the YYD motif (*Mann et al., 2012*). This suggests that phosphorylated Gogo regulates Hts enzymatic activity rather than binding. The enzymatic activity of the Hts homolog adducin is controlled by Ser/Thr kinases in mammals (*Fukata et al., 1999*; *Matsuoka et al., 1996*; *Matsuoka et al., 2000*). This type of Ser/Thr kinase activation might occur in conjunction with the activation of the Tyr kinase that phosphorylates the

Gogo YYD motif. These regulations may result in Gogo counteracting Fmi to suppress radial filopodia extension, thereby suppressing R8 axon-axon interactions during phase 2.

## Genomic economy of Gogo regulation in neuronal circuit formation

This study demonstrates that the insulin secreted from surface and cortex glia switches the phosphorylation status of Gogo, thereby regulating its two distinct functions. Non-phosphorylated Gogo mediates the initial recognition of the glial protrusion in the medulla column center. Phosphorylated Gogo suppresses radial filopodia extension by counteracting Fmi to prevent axon bundling and to maintain the one axon-to-one column ratio (*Figure 7J*).

Phosphorylated protein is typically activated or inactivated by phosphorylation. For example, to become activated and transduce downstream signaling, Robo and Eph have tyrosine phosphorylation sites and need to be dephosphorylated or phosphorylated, respectively (*Dearborn et al., 2002*; *Sun et al., 2000*). Few proteins have two distinct functions that are independently assigned to phosphorylation status (*Li et al., 2018*), and the current study demonstrates that Gogo is one of them. This mechanism is of great interest from a genomic economy point of view, where the animal genome takes an economical strategy to maximize protein functions and networks with a limited number of genes. The genomic economical strategy was likely important in the establishment of complex functional neuronal circuits during the evolution of higher-order species. Therefore, this mechanism is highly likely to be conserved across species.

# Materials and methods

## Fly strains and genetics

Flies were kept in standard *Drosophila* media at 25°C unless otherwise indicated. The following fly stocks and mutant alleles were used: sensFLP, 20C11FLP, GMR-(FRT.Stop)-Gal4 (*Chen et al., 2014*); *gogo*[H1675], *gogo*[D1600], UAS-GogoT1, ato-Δmyc, GMR-GogoΔN-D, GMR-GogoΔN-E, GMR-GogoΔN-G, GMR-GogoΔN-H, UAS-GogoFL-myc, UAS-GogoΔC-myc, UAS-GogoΔN-myc (*Tomasi et al., 2008*); UAS-GogoΔC, <*gogo*<, <*fmi*N<, *fmi*[E59], UAS-Fmi, UAS-Fmi ΔC (*Hakeda-Suzuki et al., 2011*); UAS-GogoFL-P40, UAS-GogoFFD-P40, UAS-GogoDDD-P40, GMR-GogoFFD-myc, GMR-gogoDDD-myc (*Mann et al., 2012*); UAS-*add1*-myc, *hts*[null] (*Ohler et al., 2011*); sens-lexA, LexAop-myrTomato, bshM-Gal4, UAS-myrGFP (*Trush et al., 2019*); GMR-Rho1 (*Hariharan et al., 1995*); dilp7-Gal4 (*Yang et al., 2008*); dlip4-Gal4 is a gift from Dr. Pierre-Yves Plaçais (CNRS France).

The following stocks used in this study are available in stock centers: UAS-FRT-stop-FRT-mcd8GFP, loco-Gal4, Act-Gal4, sensGal4, R85G01Gal4, R25A01Gal4, Mz97Gal4, UAS-stinger, Rh6-mCD8-4xGFP-3xmyc, Rh4-mCD8-4xGFP-3xmyc, *gogo*-Gal4, OK107-Gal4, UAS-dicer2, UAS-40D, tub-Gal80[ts], UAS-FLP, UAS-mCD8GFP, UAS-myrRFP, UAS-nlsGFP, UAS-shi[ts1], UAS-htsRNAi, UAS-hobRNAi, UAS-dlip1 RNAi, UAS-dlip2 RNAi, UAS-dlip3 RNAi, UAS-dlip4 RNAi, UAS-dlip5 RNAi, UAS-dlip6 RNAi, UAS-dlip7 RNAi, UAS-dlip8 RNAi, dlip1-Gal4, dlip2-Gal4, dlip3-Gal4, dlip5-Gal4, UAS-Fz RNAi, UAS-Fz2 RNAi, UAS-dsh RNAi, UAS-Gq RNAi, UAS-Go RNAi, UAS-GsRNAi, UAS-Gi RNAi, UAS-Gf RNAi, UAS-cta RNAi (BDSC); dilp6-Gal4 (DGRC); UAS-gogoRNAi UAS-fmiRNAi (VDRC). The following fly strains were generated in this work: *gogo*-FSF-GFP, *fmi*-FSF-mcherry, gogoΔGOGO1, gogoΔGOGO2, gogoΔGOGO3, gogoΔGOGO4, gogoΔCUB, gogoΔTSP1, gogoFlp-stop. The specific genotypes utilized in this study are listed in Table S1.

## Generation of Gogo-FsF-GFP and Fmi-FsF-GFP knock-in allele

Gogo-FsF-GFP and Fmi-FsF-GFP knock-in allele was generated by CRISPR/Cas9 technology (*Kondo and Ueda, 2013*). A knock-in vector containing the homology arms, the flip-out cassette with GFP (FRT-stop-FRT-GFP), and the red fluorescent transformation marker gene (3xP3RFP) was generated as described previously (*Trush et al., 2019*). The oligo DNAs used for amplification of Gogo and Fmi fragments and creating gDNA are listed in Supplementary file 2. A gRNA vectors were injected to eggs of yw; attP40[nos-Cas9]/CyO or y1 w1118; attP2[nos-Cas9]/TM6C, Sb Tb together with the knock-in vector. The precise integration of the knock-in vector was verified by PCR and sequencing.

The gogo mutants deleting a specific domain gogoΔGOGO1, gogoΔGOGO2, gogoΔGOGO3, gogoΔGOGO4, gogoΔCUB, and gogoΔTSP1 mutants were generated by CRISPR/Cas9 technology (*Kondo and Ueda, 2013*). A part of gogo gene deleting a specific domain was amplified by overlapping PCR. Single or multiple gDNA vectors were created and cloned into pBFv-U6.2 vector. The DNA oligos used for cloning and creating the gDNA are listed in *Supplementary file 2*.

## Generation of GogoFlpStop mutant

GogoFlpStop mutant was generated by replacing *gogo* intronic MiMIC cassette (BDSC; 61010) with the FlpStop cassette using φC31 integrase (*Hu et al., 2011*). The FlpStop cassette is a gift from Dr. Thomas R Clandinin.

## Immunohistochemistry and imaging

The experimental procedures for brain dissection, fixation, and immunostaining as well as agarose section were as described previously (*Hakeda-Suzuki et al., 2011*). The following primary antibodies were used: mAb24B10 (1:50, DSHB), rat antibody to CadN (Ex#8, 1:50, DSHB), mouse antibody to Repo (8D12, 1:20, DSHB) mouse antibody to myc (4E10, 1:100, Santa Cruz), rabbit antibody to RFP (1:500 ROCKLAND), rabbit antibody to GFP conjugated with Alexa488 (1:200, Life technologies). The secondary antibodies were Alexa488, Alexa568, or Alexa633-conjugated (1:400, Life technologies). Images were obtained with Nikon C2$^+$ and A1 confocal microscopes and processed with Adobe Photoshop and Illustrator.

Live imaging was done according to *Özel et al., 2015*. Images were obtained with Zeiss LSM880NLO + COHERENT Chameleon Vision.

## Quantitative methods

In *Figure 1C–L*, average Gogo-GFP or Fmi-mCherry intensity was calculated in each axon termini. GFP or mCherry per axon of the confocal image was manually selected by ImageJ and averaged (max. 85, n = 3, 24 axons each). Each axon was identified by R8-specific maker (myr-RFP, mCD8GFP) or staining with mAb24B10.

In *Figure 1N*, the relative fluorescence intensities of both Gogo-GFP and Fmi-mCherry labels are plotted for a representative dotted line drawn from the edge to the center of the medulla column (as shown in *Figure 1M*). Since the total fluorescence intensity of GFP, mCherry, and 24B10 stained per axon is different, each intensity was normalized by the total intensity for each axon. The histogram in *Figures 4D, E* and *7B'* were also quantified along the dotted lines.

In *Figures 2D*, *3E*, *5F*, *Figure 5—figure supplement 1H*, *Figure 7E*, the number of abnormal R8 axon terminal was calculated manually as a fraction of all GFP-expressing photoreceptors in third instar larvae (phase 1). Abnormal R8 axon terminal was defined as the termini intruded into the medulla columnar center and failed to form a proper horseshoe shape.

In *Figure 2D*, the diameter of the medulla column was measured, the longest diameter of the circular structure stained by anti-Ncad antibody.

In *Figures 2H*, *5Q*, *Figure 5—figure supplement 1B*, *Figure 6C,M*, the number of R8 axon invasions were calculated manually as a fraction of all GFP-expressing photoreceptors in 24% APF (phase 2). Since R8 axons overlapped before entering the medulla, a precise quantification was not possible and we estimated the bundling, we compared the number of R8 photoreceptors invading medulla between wild type and the tested sample.

In *Figures 2L* and *3H*, the number of R8 axons that failed to extend filopodia to medulla neuropil was calculated manually as a fraction of all GFP-expressing photoreceptors in 48% APF (phase 3). Medulla surface (M0) was identified by staining with anti-Ncad.

In *Figure 5—figure supplement 1E*, the number of R8 axons stopping at M0 was calculated manually as a fraction of all GFP-expressing photoreceptors in adult stage. Medulla surface (M0) was identified by staining with anti-Ncad.

In *Figure 3L*, *Figure 3—figure supplement 2O*, 5L, the length of the longest filopodia was measured in 3D images. The 3D images were taken by the Nikon A1 confocal microscope with a thickness of 40 μm. The 3D images were subdivided into 10 μm thicknesses and the length of filopodia was measured. The 3D reconstruction was done using Nikon NIS-Elements AR Analysis. Multiple filopodia extended from one axon, but only the longest filopodia was measured. Each axon and

filopodia can be identified by adjusting the brightness. The longest filopodia was measured from M0 using anti-Ncad staining as a reference.

## Acknowledgements

We gratefully acknowledge Dr. Pierre-Yves Plaçais (CNRS France) for providing the dlip4-Gal4 line and Dr. Thomas R Clandinin (Stanford Univ.) for FlpStop cassette. We thank Kyoto Stock Center (DGRC), Bloomington *Drosophila* Stock Center, Vienna *Drosophila* Resource Center (VDRC), and Developmental Studies Hybridoma Bank for providing fly or antibody stocks. We thank Enago (http://www.enago.jp) for the English language review. This work was supported by Grant-in-Aid for JSPS Fellows 19J14499 (HT), 18J00367 (YN), JSPS KAKENHI Grant number 18K06250 (SH-S), 18K14835 (YN), 17H03542 (MS), 17H04983 (AS), 19K22592 (AS), Grant-in Scientific Research on Innovation Areas from the Ministry of Education, Culture, Sports, Science, and Technology of Japan 'Dynamic regulation of Brain Function by Scrap and Build System' 16H06457 (TS), 17H05739 (MS) 'Interplay of developmental clock and extracellular environment in brain formation' 17H05761 (MS), 19H04771 (MS), Takeda Science Foundation Life Science Research Grant (AS) and Takeda Visionary Research Grant from the Takeda Science Foundation (TS).

## Additional information

### Funding

| Funder | Grant reference number | Author |
|---|---|---|
| Japan Society for the Promotion of Science | 19J14499 | Hiroki Takechi |
| Japan Society for the Promotion of Science | 18J00367 | Yohei Nitta |
| Japan Society for the Promotion of Science | 18K06250 | Satoko Hakeda-Suzuki |
| Japan Society for the Promotion of Science | 18K14835 | Yohei Nitta |
| Japan Society for the Promotion of Science | 17H03542 | Makoto Sato |
| Japan Society for the Promotion of Science | 17H04983 | Atsushi Sugie |
| Japan Society for the Promotion of Science | 19K22592 | Atsushi Sugie |
| Ministry of Education, Culture, Sports, Science and Technology | 16H06457 | Takashi Suzuki |
| Ministry of Education, Culture, Sports, Science and Technology | 17H05739 | Makoto Sato |
| Ministry of Education, Culture, Sports, Science and Technology | 17H05761 | Makoto Sato |
| Ministry of Education, Culture, Sports, Science and Technology | 19H04771 | Makoto Sato |
| Takeda Science Foundation | Life Science Research Grant | Atsushi Sugie |
| Takeda Science Foundation | Visionary Research Grant | Takashi Suzuki |

The authors declare that there was no funding for this work.

## Author contributions
Hiroki Takechi, Conceptualization, Data curation, Formal analysis, Validation, Investigation, Writing - original draft, Project administration, Writing - review and editing; Satoko Hakeda-Suzuki, Conceptualization, Formal analysis, Funding acquisition, Investigation, Writing - original draft, Writing - review and editing; Yohei Nitta, Conceptualization, Funding acquisition, Investigation, Visualization, Writing - original draft, Writing - review and editing; Yuichi Ishiwata, Riku Iwanaga, Formal analysis, Investigation; Makoto Sato, Resources, Supervision, Funding acquisition; Atsushi Sugie, Conceptualization, Supervision, Funding acquisition, Writing - review and editing; Takashi Suzuki, Conceptualization, Resources, Supervision, Funding acquisition, Investigation, Writing - original draft, Project administration, Writing - review and editing

## Author ORCIDs
Satoko Hakeda-Suzuki  https://orcid.org/0000-0001-8749-1479
Yohei Nitta  http://orcid.org/0000-0002-0712-428X
Makoto Sato  http://orcid.org/0000-0002-7763-0751
Takashi Suzuki  https://orcid.org/0000-0001-9093-2562

## Decision letter and Author response
Decision letter https://doi.org/10.7554/eLife.66718.sa1
Author response https://doi.org/10.7554/eLife.66718.sa2

# Additional files

## Supplementary files
- Supplementary file 1. List of genotypes used.
- Supplementary file 2. oligo DNAs used for generating and analyzing transgenic flies.

## Data availability
All data generated or analysed during this study are included in the manuscript and supporting files. Source data files have been provided.

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
