## [Decision Letter]

**Acceptance summary:**

This work contributes to our understanding of how neurons and glia interact during *Drosophila* optic lobe development. It makes the remarkable finding that the cell adhesion protein, Gogo, that interacts with an atypical cadherin, Fmi, switches from cooperating with Fmi to opposing it during development. This switch in function appears to be controlled by Gogo phosphorylation by the insulin receptor in response to a signal from glial cells.

**Decision letter after peer review:**

[Editors’ note: the authors submitted for reconsideration following the decision after peer review. What follows is the decision letter after the first round of review.]

Thank you for submitting your work entitled "Glial insulin regulates cooperative or antagonistic Golden goal/Flamingo interactions during photoreceptor axon guidance" for consideration by *eLife*. Your article has been reviewed by a Senior Editor, a Reviewing Editor, and three reviewers. The following individuals involved in review of your submission have agreed to reveal their identity: Thomas Hummel (Reviewer #1); Franck Pichaud (Reviewer #3).

Our decision has been reached after consultation between the reviewers. As you know, this is a critical and unique part of the *eLife* process and allows the reviewers to discuss their opinion, to realize things they might have missed or to correct opinions. Based on an intense discussion on your paper and on the individual reviews below, we regret to inform you that your work will not be considered further for publication in *eLife*.

The reviewers did find the topic to be of significant interest but several serious issues preclude publication in *eLife*. However, if you were able on the future to address these concerns, we would be happy to receive a new submission that could be processed fairly quickly.

The main issues are as follows:

- The evidence supporting a switch in function for Gogo with respect to Fmi is not strong enough. The main evidence is based on genetic interactions and it is not clear that the three step mechanism can explain pathfinding.

The reviewers were all concerned about step 2 and would have liked to see better arguments to support and explain what this step two is, and how one specific step is affected by another.

- Although the role of Insulin-induced phopshorylation and that fact that Gogo function is regulated by the insulin pathway are an exciting phenomenon, this was already published in your own paper in 2012 and is thus not totally new, even though the source of the signal was now identified.

- The paper lacks quantification and some of the manipulations that are not specific enough to lead to strong conclusions (eg hobbit).

Reviewer #1:

The study by Suzuki and colleagues addresses the developmental assembly of afferent axons into columnar circuit units, a highly conserved structural feature throughout nervous systems. Previous work has identified a close functional interaction between the two transmembrane proteins FMI and GOGO in *Drosophila* columnar photoreceptor targeting and the current study provides novel insights in the temporal dynamics and cellular interactions of this molecular cross talk. Based on gene expression analysis, genetic interaction studies and cell type specific manipulations the authors could show that GOGO modulates FMI activity in specific developmental steps and that glial cells play in important role in neuronal GOGO-FMI modulation.

The manuscript is well written and the presented data are of high quality. The generation of novel sophisticated genetic tools allows a detailed analysis of the sequential cellular interaction and provides a new level of understanding about the complexity and hierarchy of neuronal connectivity development. However, regarding the proposed mechanism, the manuscript could be more convincing by addressing some issues regarding the function of GOGO in the second step of axon targeting and the role of insulin signaling and FMI adhesion in axon-glia interaction.

While the localization of the synergistic role of GOGO in supporting FMI adhesion is convincing and in is in line of what has been demonstrated before, the proposed subsequent antagonistic activity seems less clear. First of all, regarding the subdivision of R cell targeting into 3 consecutive steps, step1 – axon targeting – and step 3 – axon vertical extension – are well justified as distinct morphological processes but what is happening at step 2? The manuscript makes different statements about step 2, like R8 axons remain at the medulla neuropil surface in the introduction or proper filopodia extension in the phenotype description. The failure to suppress inappropriate horizontal filopodia extensions in GOGO mutants seems a direct consequence of the axon-glia adhesion defects in step 1, which later result in multi-axon columns. Therefore if step 2 is better defined as a developmental process, the importance of the proposed antagonistic GOGO function will become more logical. Alternatively, without step 2 a simplified model with a singular function of GOGO in supporting FMI-mediated axon-glia interaction at step 1, which is then followed by the GOGO downregulation due to glia-induced DILR signaling seems equally likely. In addition, as there is no GOGO expression at step3, it doesn't make sense to investigate the role of Gogo during step3 or conclude that Gogo and Fmi function in opposing manners during step 3 of R8 axon targeting. The fact that prolonged GOGO expression stabilizes the initial axon targeting step is in lines with a mechanisms which simply removes GOGO from the FMI-mediated axon-glia contact thereby allowing further axon extension.

Regarding the functional activity of GOGO at step 2, the main data supporting a switch from cooperation into an antogonistic function are derived from different sets of genetic interaction studies, with double knock down are only analyzed for step 1 indicating synergistic interactions and combined OE and KD are only shown for step 2 concluding antagonistic effects of GOGO on FMI. To exclude a different effect of the genetic manipulation and further support of a stage-specific change in GOGO activity, the modulation of the FMI GOF phenotype should be analyzed in step 1 and the single gene and double knock down could be targeted to stage 2 using the Gal80ts method. Similarly, in the localization of phosphorylated GOGO, a direct comparison of mutant rescue in step 1 and GOF modification for step 2 is problematic to conclude a differential requirement. Here the same experiments should be conducted for all developmental time points. In addition, as there is no direct result provided that the phosphorylation status of GOGO changes in step2, this central statement by the authors is only built on the genetic interaction studies and needs additional experimental data.

The role of insulin signaling by glial cells in controlling axon targeting is a key aspect of the manuscript and clearly the main conceptual novelty. Covered by the last two paragraphs, the analysis largely relies on phenotypic similarities and no functional interactions. More importantly, the fact that no candidate signal could be identified makes the main conclusion regarding the non-autonomous regulation problematic at this point and should be analyzed further.

As glial cells support R cell axon guidance from the initial outgrowth all the way into the optic lobes, the R8 axons connectivity phenotypes induced by the different genetic manipulation of the insulin pathway and FMI adhesion might be an indirect consequence of earlier disturbances, like the disruption of the precise temporal ingrowth or selective axon fasciculation. Here, a more in-depth developmental analysis would be further strengthened the main conclusion of the manuscript.

First, for all gene disruption experiments and glia cell manipulation, phenotype analysis should be extended to late third instar and early pupal stages, to support the sequential activity of FMI and Insulin signaling in axon columnar connectivity. In addition, experiments could be targeted specifically to the pupal stage leaving the initial axon targeting unaffected. Second, as little is known about the role of glial cells at this stage, it would be very helpful to provide more insights into the developmental profile of glial cell morphology and the sequence of axon glia interactions. When do glial cell processes enter the medulla column? What is the spatial organization of growth cone and glial processes. How dynamic is glial cell morphology from initial axon targeting to vertical extension? Second, to further test for putative redundant dilp gene function, combined gene knock down experiments and genetic interaction studies, e.g. in an GOGO-FMI sensitized background could be performed. Are dilp mutants available?

Reviewer #2:

This is an interesting story. It shows that Gogo works with Fmi in some contexts, and against it in others. That may be regulated by Gogo phosphorylation. I am concerned, however, that the interpretation is so specific, in that they infer the phosphorylation state of Gogo in vivo based on rescue with FF and DD mutants. DD mutants are not necessarily phosphomimetic, and FF mutants are not just unphosphorylated. If a PTB protein normally interacts with the unphosphorylated tyrosine,(s) then the FF mutants would block that binding. In reality, Gogo phosphorylation in vivo is probably not that clear-cut. Also, kinases other than InR might be involved.

1) One major problem is the lack of quantitation of much of the data. This is an interesting story, but without more quantitation it is hard to understand the penetrance of the phenotypes.

2) WT and heterozygote controls are missing in many graphs and image panels. Since this paper is heavily dependent on RNAi experiments, authors need to report controls without driver. These are missing here. "wt" is not a control for RNAi; ideally, one needs to examine RNAi with no driver, and driver with no RNAi.

3) The phenotype that is reported in the paper at 24h APF, the authors refer to it as the "axon bundling" phenotype. I would refer to the phenotype at 24h APF as premature filopodial extension, which can be easily distinguished in these images. In WT, R8 do not extend filopodia till after 36h APF and they do not stabilize and elongate till after 46h APF (Orkun and Zipursky, 2016). If there is "axon bundling" (which is not visible in the current figures) a higher magnification would help show that phenotype in detail.

4) The phenotype identified in L3 as "correct axon targeting" (or "mistargeting at column" in Figure 3 graph) should be defined as R8 terminal morphology, which can be either horseshoe-shaped or oval morphology.

a) Authors should also show the N-Cadherin panel so that readers can see what the cross-sections of the columns look like in the different genotypes.

b) They should quantitate this, maybe by using the diameter of the column to normalize for the various genetic combinations.

c) Is it possible that in some of the genotypes, the cross-section of the column is smaller/constricted as compared to that in wild-type, and this affects morphology of R8 and thus an indirect effect? For eg. In Figure 5—figure supplement panel F: authors refer to fmi- + fmiρC as non-rescue, where I think there is rescue. Increase in the gain of the green R cell channel obscures the horse-shoe shape of R8 and thus looks like there is no rescue when actually there is (compare the 1st and the 3rd panel in this set).

5) M1 and M3 layers should be marked in all image panels.

6) Instead of referring as step1, 2 etc – it is hard to follow the various steps especially since the main data points are at L3, 24h APF and 48h APF. One should use the phenotype in referring to those steps:

L3 – R8 terminal morphology (horse-shoe).

24h APF – No filopodial extension.

48h APF – R8 axon terminals go to the M3 layer.

This would make it easier to interpret the phenotypes presented. For e.g. 24h APF phenotype would be premature extension and 48h APF R8 axons stopped in M1 layer or overshoot the M3 layer.

Figure 2:

Phenotypes need to be quantitated, especially since this is a new type of mutant (R8-RNAi + het for a gogo mutation). There are 3 types of phenotypes: 3rd instar column targeting-need to know % penetrance; filopodial extension in E-need % of columns affected; axon crossovers in F and failure to extend to M3 in J-need % of columns affected.

Figure 3:

a) Panel E-H: M1 and M3 layers should be marked. Don't understand what is happening in H. Do those axons stop at M3? If Gogo OE gives an Fmi mutant like phenotype with stopping at M1, shouldn't Fmi+Gogo OE give the same? Is Fmi OE overriding an inhibition of Fmi by Gogo?

b) Panel I: Include +Gogo alone and wild-type control in the quantitation. Don't understand this quantitation. What is being measured in H? I don't see any filopodia there.

Figure 4:

a) Panels A-C: Gogo localization in fmi overexpression is changed according to the authors. But is this because the morphology of R8 terminal is different from control (if one compares only the R8 panels)? Also, spacing of R8 is not present in a consistent grid like pattern that is seen in wild-type. It seems like fmi overexpression might alter R8 morphology resulting in altered Gogo localization as a secondary consequence.

b) Panels A-H: Gogo and Fmi localization in the mutants vs overexpression in L3: it is hard to understand when the authors call Fmi/Gogo is mislocalized vs unaltered b/c there is no quantitation or obvious differences to make that determination.

c) Panels I-J: Since there is an increase in filopodial extension at 24h APF in Fmi overexpression only in R8 axons, authors should quantitate that R8 overexpression phenotype (as measured by % columns with Gogo-containing filopodia going past correct layer.

Figure 5:

a) Panels A-F: This is informative data regarding structure-function analysis of Gogo protein in defining R8 morphology in L3, but WT control is missing in panel F graph (and should be shown for reference).

b) Panels G-L: Effect on R8 premature filopodial extension at 24h APF when Gogo variants are overexpressed in all PRs: Interesting data, but needs wild-type control.

c) Panels M-P: do these get fixed at later developmental stages in gogo mutant? No information in Materials and methods on how they were quantitated.

d) Figure 5—figure supplement 1: Quantitation missing on hts mutant @ 24h APF, need to compare with gogo and double mutants.

e) Figure 5—figure supplement 1: No mention of panels E-G in the text.

Figure 6:

a) Panel A: Quantitation of dinr mutant needed.

b) Panel B: Authors should split the image to show the repo+ cells only. Because one cannot see the Repo+ red signal in the 2 cells (indicated by the top arrowheads).

c) Panels C-J: It would be helpful to know when these drivers come on.Could be differences in effects due to early vs. late expression in glia.

d) hob RNAi quantitation should be included.

Figure 7:

a) Panel A: show with 24B10 where the R7 layer is, so that one can see the position of the endings of the glial process that seem to be positioned below the R8 layer.

b) Panel B: quantitate co-localization with fmi and glial marker. This is very hard to interpret.

c) Panels C-H: No quantitation shown. Difficult to see what the authors are trying to show here. Also, the authors should show horizontal orientation for glial specific knockdown of fmi in mid-pupae or adult.

d) Panel D: Columns are disorganized in glial specific knockdown of fmi. So, R8 positioning could be a result of the columnar disorganization rather than fmi instructing R8.

Reviewer #3:

I am reviewing this paper with the view that it is probably not an option to perform experiments at this time, while still making sure that the authors conclusions are well supported by the data presented in the paper. This is why I do not suggest additional experiments..

This is an interesting study by a group that are experts in studying how the transmembrane proteins Gogo and Fmi regulate targeting of the R8 axon in the fly visual system. The present work builds upon previous studies from the lab. Overall the data quality is excellent. The key observation here is that the phosphorylation status of Gogo seems to act as a switch and is required during distinct phases of R8 projection. Secretion from glial cells, and also Fmi expression in these cells, is required for proper R8 projection. These observations are interesting. However, the study suffers from significant draw backs that unfortunately prevent me from supporting it for publication at this stage.

Firstly, this study presents significant overlap with a previous paper from the lab (Mann, 2012) that reported Gogo phosphorylation/dephosphorylation regulation by the insulin pathway, and showed that this is required for proper R8 projection in the fly visual system. While the present work looks at how these phenotypes develop through time, the idea that phosphorylation of Gogo might be regulated by Dilps and that it might change Gogo's function during R8 pathfinding is not new.

I also think that there is a need to better define and present the evidence for a switch in function for Gogo with respect to Fmi. Similarly, the authors should also make it clearer the extent to which Steps 1-3 are interdependent (or not). Isn't it possible that if step 1 fails, then 2 and 3 will also fail as a consequence? This is important because the authors tend to present these steps as relatively independent, but it is not made clear why they hold that view.

The work also feels very qualitative, especially Figure 1, and Figure 4. The authors conclusion would be greatly strengthened by providing quantifications next to the panels. I feel that given this is a relatively well-studied model system, the authors should try to come up with a better way to quantify the shape and location of the R8 termini. This would make it convincing and could even reveal new interesting features of the phenotype (and wild type).

---

## [Author Response]

[Editors’ note: the authors resubmitted a revised version of the paper for consideration. What follows is the authors’ response to the first round of review.]

Reviewer #1:The study by Suzuki and colleagues addresses the developmental assembly of afferent axons into columnar circuit units, a highly conserved structural feature throughout nervous systems. Previous work has identified a close functional interaction between the two transmembrane proteins FMI and GOGO in Drosophila columnar photoreceptor targeting and the current study provides novel insights in the temporal dynamics and cellular interactions of this molecular cross talk. Based on gene expression analysis, genetic interaction studies and cell type specific manipulations the authors could show that GOGO modulates FMI activity in specific developmental steps and that glial cells play in important role in neuronal GOGO-FMI modulation.The manuscript is well written and the presented data are of high quality. The generation of novel sophisticated genetic tools allows a detailed analysis of the sequential cellular interaction and provides a new level of understanding about the complexity and hierarchy of neuronal connectivity development. However, regarding the proposed mechanism, the manuscript could be more convincing by addressing some issues regarding the function of GOGO in the second step of axon targeting and the role of insulin signaling and FMI adhesion in axon-glia interaction.While the localization of the synergistic role of GOGO in supporting FMI adhesion is convincing and in is in line of what has been demonstrated before, the proposed subsequent antagonistic activity seems less clear. First of all, regarding the subdivision of R cell targeting into 3 consecutive steps, step1 – axon targeting – and step 3 – axon vertical extension – are well justified as distinct morphological processes but what is happening at step 2? The manuscript makes different statements about step 2, like R8 axons remain at the medulla neuropil surface in the introduction or proper filopodia extension in the phenotype description. The failure to suppress inappropriate horizontal filopodia extensions in GOGO mutants seems a direct consequence of the axon-glia adhesion defects in step 1, which later result in multi-axon columns. Therefore if step 2 is better defined as a developmental process, the importance of the proposed antagonistic GOGO function will become more logical. Alternatively, without step 2 a simplified model with a singular function of GOGO in supporting FMI-mediated axon-glia interaction at step 1, which is then followed by the GOGO downregulation due to glia-induced DILR signaling seems equally likely. In addition, as there is no GOGO expression at step3, it doesn't make sense to investigate the role of Gogo during step3 or conclude that Gogo and Fmi function in opposing manners during step 3 of R8 axon targeting. The fact that prolonged GOGO expression stabilizes the initial axon targeting step is in lines with a mechanisms which simply removes GOGO from the FMI-mediated axon-glia contact thereby allowing further axon extension.

We thank the reviewer for their comments. While we fully understand the reviewer's point, we still believe that step (phase) 2 exists, and that it is required for normal R8 axon targeting. For that, we have the following 4 lines of evidence:

1) Glial *fmi* loss-of-function mutation disrupted the horseshoe shape at phase 1, but not bundling at phase 2. We have added this data to Figure 7C’ and D’.

2) The nonphosphorylated GogoFFD rescue was not perfect, and there was still some axonal bundling, shown in Figure 5O.

3) We also added an experiment as suggested below by the reviewer: the phase 2-specific *gogo* RNAi knockdown experiment using Gal80^ts^. As we anticipated, the *gogo* LOF phenotype characterized by longer filopodia in more radial directions, was also observed in the phase 2-specific *gogo* knockdown, suggesting that the defects of Gogo LOF in phase 2 are independent of phase 1. We described these results in Figure 2O-P''.

4) The DInR LOF mutant does not have defects in phase 1 (horseshoe shape), but exhibits a bundling phenotype in phase 2 (Figure 6A and Figure 6B, respectively). Therefore, there exists a signaling molecule that specifically regulates only phase 2, but not phase 1.

Regarding the functional activity of GOGO at step 2, the main data supporting a switch from cooperation into an antogonistic function are derived from different sets of genetic interaction studies, with double knock down are only analyzed for step 1 indicating synergistic interactions and combined OE and KD are only shown for step 2 concluding antagonistic effects of GOGO on FMI.

We have already included "combined OE and KD in phase 1*”* in Figure 3-figure supplement 1I-L).

Images of phase 2 in the double knockdown mutants have also been added to Figure 3-figure supplement 1H). Here we demonstrate that *fmi* LOF can suppress the bundling phenotype of *gogo* LOF in phase 2.

To exclude a different effect of the genetic manipulation and further support of a stage-specific change in GOGO activity, the modulation of the FMI GOF phenotype should be analyzed in step 1.

These data were included in Figure 3—figure supplement 1I and K in the previous version. Here we showed that, during phase 1, Fmi OE & *gogo* LOF showed much greater filopodia extension with a disrupted horseshoe shape compared to mere Fmi OE. We believe the reason why the filopodia are affected even in phase 1 is because the function of *gogo* cannot be separated perfectly with phases or time, but rather the shift may happen gradually, and species in either phase might coexist or be loosely separated in a spatial manner (e.g. from the growth cone rim at the column center to the filopodia towards outside) at certain time points between phase 1 and phase 2. We discuss this issue in the Discussion section.

…and the single gene and double knock down could be targeted to stage 2 using the Gal80ts method.

We thank the reviewer for their helpful suggestions. As mentioned above, we have addressed the question of “what is step 2? " by adding additional characterization data. We added an experiment conducting a stage-specific *gogo* knockdown experiment using Gal80^ts^. As we anticipated, the *gogo* LOF phenotype characterized by longer filopodia in more radial directions was also observed in step 2-specific *gogo* knockdowns, suggesting that step 2 is independent of step 1. These results are displayed in Figure 2O-P''.

We observed that adding *fmi* LOF suppressed the *gogo* LOF bundling phenotype, as shown in Figure 3—figure supplement 1F-H. We could not conduct double LOF Gal80^ts^ experiments as it was prohibitively highly technically demanding to transfect Gal80^ts^ on top of a double LOF. We believe that it is sufficient to show the antagonistic effect in phase 2 in double LOF mutants.

Similarly, in the localization of phosphorylated GOGO, a direct comparison of mutant rescue in step 1 and GOF modification for step 2 is problematic to conclude a differential requirement. Here the same experiments should be conducted for all developmental time points.

We have added the DDD rescue in phase 2 to Figure 5P, and FFD DDD OE in phase 1 to Figure 5—figure supplement 1F in the revised manuscript.

In addition, as there is no direct result provided that the phosphorylation status of GOGO changes in step2, this central statement by the authors is only built on the genetic interaction studies and needs additional experimental data.

We agree with the reviewer that there is no direct result showing the change in phosphorylation status in phase 2. In spite of our efforts to generate phospho-specific Gogo antibodies, we could not observe staining in tissue samples. We only could show that the phosphorylated form existed in larval eye discs, as in our previous paper (Mann et al., 2012). As we argued in the Discussion section, the change in phosphorylation status can be very slow, or does not necessarily even happen, as spatial separation throughout the phases can also theoretically achieve the proper R8 targeting.

Similarly, when we overexpressed Gogo in phase 1 in a *fmi* LOF background, we observed filopodia suppression, which is the Gogo function normally seen in phase 1 (Figure 3—figure supplement 1J-L). Again, we believe that the function of Gogo cannot be separated perfectly by phase or by time, but rather we think that the shift happens gradually, and both species may co-exist at certain time points. We discuss this issue further in the Discussion section.

The role of insulin signaling by glial cells in controlling axon targeting is a key aspect of the manuscript and clearly the main conceptual novelty. Covered by the last two paragraphs, the analysis largely relies on phenotypic similarities and no functional interactions. More importantly, the fact that no candidate signal could be identified makes the main conclusion regarding the non-autonomous regulation problematic at this point and should be analyzed further.

We determined a genetic interaction between *dilp6* and *fmi* in a glia-specific manner, indicating that *dilp6* is at least indirectly involved in this signal. The details of the experiments are explained below, and the additional data has been added to Figure 6N-Q.

As glial cells support R cell axon guidance from the initial outgrowth all the way into the optic lobes, the R8 axons connectivity phenotypes induced by the different genetic manipulation of the insulin pathway and FMI adhesion might be an indirect consequence of earlier disturbances, like the disruption of the precise temporal ingrowth or selective axon fasciculation. Here, a more in-depth developmental analysis would be further strengthened the main conclusion of the manuscript.First, for all gene disruption experiments and glia cell manipulation, phenotype analysis should be extended to late third instar and early pupal stages, to support the sequential activity of FMI and Insulin signaling in axon columnar connectivity. In addition, experiments could be targeted specifically to the pupal stage leaving the initial axon targeting unaffected.

First, we would like to point out that the glia that guide R8 axons to the center of the column (Figure 7B, C; phase 1) and the glia that mediate insulin secretion (Figure 6—figure supplement 1H and I; phase 1- 2) are different population of cells. Therefore, we respectfully disagree with the reviewer that the initial defect in glia population 1 can result in the defects in glia population 2.

To clarify this point, as pointed out by the reviewer, we have added images of glial-specific *fmi* LOF mutants at APF24% and DInR R8 LOF mutants at third instar (horseshoe morphology) to Figures6 and Figure 7.

Glial-specific *fmi* LOF mutation disrupted the horseshoe shape in phase 1, but phase 2 appeared perfectly normal, while DInR R8 LOF mutants at the third instar showed normal phase 1 but disrupted phase 2 morphology (bundling). Since these two genes (*fmi* and InR) showed very different LOF phenotypes in different phases, we do not believe that the first defect (Fmi) can be the source of the other defects observed later in the system.

Second, as little is known about the role of glial cells at this stage, it would be very helpful to provide more insights into the developmental profile of glial cell morphology and the sequence of axon glia interactions. When do glial cell processes enter the medulla column? What is the spatial organization of growth cone and glial processes. How dynamic is glial cell morphology from initial axon targeting to vertical extension?

We thank the reviewer for this encouraging comment. We have shown the sequential organization in Figure 7A, since the medulla contains a relatively wide span of R8 axons of different ages. We also added the text below to explain when the glial process enters the column in the Results section. In the oldest glia column, the glia extension appears to have begun retracting (Figure 7A, yellow arrow). We also added an image of APF24% Loco-Gal4 UAS-mCD8::GFP to observe the development of these glia. The column-center glia were retracted at least to the M1 layer (this description was added to the Results section). After that time point, we are not confident about what happens to the column-center glia, due to the lack of specific promoter drivers to follow.

We added the following text to the result section:

“The glial protrusion seems to extend into the medulla layers as early as the R8 growth cone enters (arrowhead in Figure 7A). The protrusion passes the R8 growth cone and extends deeply to the medulla layers. However, it starts to retracts towards the latest period of third instar larvae (yellow arrow in Figure 7A), and completely retracted from the medullar layers in APF24% (phase 2) (Figure 7C’).”

Second, to further test for putative redundant dilp gene function, combined gene knock down experiments and genetic interaction studies, e.g. in an GOGO-FMI sensitized background could be performed. Are dilp mutants available?

As mentioned in the text, the *dilp6* mutant alone does not display any defects in R8 axon targeting. However, thanks to the reviewer's suggestion, we performed additional experiments in a sensitized background and looked for genetic interactions. To overcome the redundancy of *dilp* function, we utilized the Fmi OE as a sensitized background to explore GOGO-FMI function.

Fmi OE already demonstrated extended filopodia, counteracting the putative phosphorylated Gogo function. In this situation, if *dilp* signal gets weaker, the phosphorylated Gogo should be reduced and will have more filopodia extension. To test this hypothesis, we knocked down *dilp6* only in glial cells in the Fmi OE background. We observed a significant enhancement in filopodia extension compared to the control, indicating that *dilp6* from glial cells likely affects filopodia extension through GOGO phosphorylation.

The additional data has been added to Figure 6 as N-Q.

Reviewer #2:This is an interesting story. It shows that Gogo works with Fmi in some contexts, and against it in others. That may be regulated by Gogo phosphorylation. I am concerned, however, that the interpretation is so specific, in that they infer the phosphorylation state of Gogo in vivo based on rescue with FF and DD mutants. DD mutants are not necessarily phosphomimetic, and FF mutants are not just unphosphorylated. If a PTB protein normally interacts with the unphosphorylated tyrosine,(s) then the FF mutants would block that binding. In reality, Gogo phosphorylation in vivo is probably not that clear-cut. Also, kinases other than InR might be involved.1) One major problem is the lack of quantitation of much of the data. This is an interesting story, but without more quantitation it is hard to understand the penetrance of the phenotypes.

We thank the reviewer for their comments. We have quantified much of the data, now shown in Figure 1C-L and N, Figure 2D, H, and L, Figure 3H, Figure 4D and E, Figure 6C and Q, Figure 7B’ and E, and Figure 3-figure supplement 1I and Figure 5—figure supplement 1B and H.

2) WT and heterozygote controls are missing in many graphs and image panels. Since this paper is heavily dependent on RNAi experiments, authors need to report controls without driver. These are missing here. "wt" is not a control for RNAi; ideally, one needs to examine RNAi with no driver, and driver with no RNAi.

We agree with the reviewer about the need for controls; however, because we are using R8-Gal4 to drive UAS-GFP to observe the shape of the R8 growth cone, taking out R8-Gal4 is not technically possible. The "WT" indicated here is actually Gal4 with no RNAi, which we believe to be the best control possible.

3) The phenotype that is reported in the paper at 24h APF, the authors refer to it as the "axon bundling" phenotype. I would refer to the phenotype at 24h APF as premature filopodial extension, which can be easily distinguished in these images. In WT, R8 do not extend filopodia till after 36h APF and they do not stabilize and elongate till after 46h APF (Orkun and Zipursky, 2016). If there is "axon bundling" (which is not visible in the current figures) a higher magnification would help show that phenotype in detail.

We agree with the reviewer, and have changed the figure titles to read “ratio of R8 axon invasion” for all of these phenotypes. We also observed that the R8 axon terminals could not line up at the M0 layer at APF24% (phase 2), and invaded the medulla layers. Since this type of invasion generally coincides with axonal bundling and disorganization of the growth cone distances, we hypothesized that the invasion was likely to be an indirect consequence of the disorganization and bundling of R8 axons, which is consistent with our finding that in the *gogo* mutant, filopodia extension could not be suppressed.

We have added magnified images to Figure 2E’-G’, and also changed the titles of the quantification graphs.

4) The phenotype identified in L3 as "correct axon targeting" (or "mistargeting at column" in Figure 3 graph) should be defined as R8 terminal morphology, which can be either horseshoe-shaped or oval morphology.

We have adopted this change to “R8 axon terminal morphology”, as suggested.

a) Authors should also show the N-Cadherin panel so that readers can see what the cross-sections of the columns look like in the different genotypes.

We have added these panels to Figure 2A-C and Figure 3A-D.

b) They should quantitate this, maybe by using the diameter of the column to normalize for the various genetic combinations.

We have quantified the diameter of the columns of *gogo* or *fmi* LOF mutants, shown in Figure 2A-C. The column diameters of the *gogo* or *fmi* LOF mutants were all similar, indicating that the targeting phenotype is not due to the column size.

c) Is it possible that in some of the genotypes, the cross-section of the column is smaller/constricted as compared to that in wild-type, and this affects morphology of R8 and thus an indirect effect? For eg. In Figure 5—figure supplement panel F: authors refer to fmi- + fmiρC as non-rescue, where I think there is rescue. Increase in the gain of the green R cell channel obscures the horse-shoe shape of R8 and thus looks like there is no rescue when actually there is (compare the 1st and the 3rd panel in this set).

We agree that *fmi- + fmi*D*C* looks different than the control. This is because fmiD*C* can induce filopodia extension, as indicated in Figure 5—figure supplement 1H. We also found that the filopodial structure in the growth cone is more enhanced than the control; however the horseshoe shape is almost completely absent in *fmi- + fmi*D*C*, and there was no significant difference compared to the control (Figure 5—figure supplement 1H,G).

5) M1 and M3 layers should be marked in all image panels.

We have added these lines as suggested by the reviewer.

6) Instead of referring as step1, 2 etc – it is hard to follow the various steps especially since the main data points are at L3, 24h APF and 48h APF. One should use the phenotype in referring to those steps:L3 – R8 terminal morphology (horse-shoe).24h APF – No filopodial extension.48h APF – R8 axon terminals go to the M3 layer.This would make it easier to interpret the phenotypes presented. For e.g. 24h APF phenotype would be premature extension and 48h APF R8 axons stopped in M1 layer or overshoot the M3 layer.

According to the suggestion, we have changed the text accordingly:

Phase 1: Gogo function 1: L3- R8 terminal morphology (horseshoe)

Phase 2: Gogo function 2: 24% APF – No bundling

Phase 3: no Gogo function: 48% APF – R8 axon extend to the M3 layer

Figure 2:Phenotypes need to be quantitated, especially since this is a new type of mutant (R8-RNAi + het for a gogo mutation). There are 3 types of phenotypes: 3rd instar column targeting-need to know % penetrance; filopodial extension in E-need % of columns affected; axon crossovers in F and failure to extend to M3 in J-need % of columns affected.

We have added quantification in Figure 2D, H, and L.

Figure 3:a) Panel E-H: M1 and M3 layers should be marked. Don't understand what is happening in H. Do those axons stop at M3? If Gogo OE gives an Fmi mutant like phenotype with stopping at M1, shouldn't Fmi+Gogo OE give the same? Is Fmi OE overriding an inhibition of Fmi by Gogo?

We have marked the temporary layer with magenta line in the revised manuscript.

Yes, Fmi OE partially overrides the Gogo OE phenotype, which we believe is an additive effect.

b) Panel I: Include +Gogo alone and wild-type control in the quantitation. Don't understand this quantitation. What is being measured in H? I don't see any filopodia there.

In this experiment, we have added +Gogo alone and wild-type control to the quantitation. Short filopodia can be seen in Panel H (now I in the revised figure).

Figure 4:a) Panels A-C: Gogo localization in fmi overexpression is changed according to the authors. But is this because the morphology of R8 terminal is different from control (if one compares only the R8 panels)? Also, spacing of R8 is not present in a consistent grid like pattern that is seen in wild-type. It seems like fmi overexpression might alter R8 morphology resulting in altered Gogo localization as a secondary consequence.

We have shown that Gogo is localized to the stalk upon Fmi OE (Figure 4D,E). The growth cone morphology does not change much, suggesting that the mislocalization is not due to the growth cone morphology. We have quantified the data in a graph to be more precise (Figure 4D,E).

b) Panels A-H: Gogo and Fmi localization in the mutants vs overexpression in L3: it is hard to understand when the authors call Fmi/Gogo is mislocalized vs unaltered b/c there is no quantitation or obvious differences to make that determination.

We agree with the reviewer that Gogo and Fmi localization is difficult to compare, especially in the mutants since the growth cone morphology is heavily altered. We have changed the text as follows to emphasize only the OE situation.

“In the LOF mutants, it was not possible to interpret the changes in localization, since the growth cone morphology had changed drastically. Therefore, we focused on situations in which the proteins were overexpressed.”

For the OE situation, we also added a visualization of the measurement of the localization signals of Gogo (Figure 4D,E).

c) Panels I-J: Since there is an increase in filopodial extension at 24h APF in Fmi overexpression only in R8 axons, authors should quantitate that R8 overexpression phenotype (as measured by % columns with Gogo-containing filopodia going past correct layer.

We have quantified the filopodial extension phenotype elsewhere (Figure 3I and L); in this panel, we only sought to highlight the mislocalization of Gogo protein in Fmi OE background.

Figure 5:a) Panels A-F: This is informative data regarding structure-function analysis of Gogo protein in defining R8 morphology in L3, but WT control is missing in panel F graph (and should be shown for reference).

We have added the WT to the quantification graph (Figure 5F).

b) Panels G-L: Effect on R8 premature filopodial extension at 24h APF when Gogo variants are overexpressed in all PRs: Interesting data, but needs wild-type control.

We have added the WT to the quantification graph (Figure 5L).

c) Panels M-P: do these get fixed at later developmental stages in gogo mutant? No information in Materials and methods on how they were quantitated.

The bundling phenotype seems to be less drastic in adult stages, but still not perfect, as we showed in our previous study (Mann et al., 2012). We have added an explanation to the Materials and methods section.

d) Figure 5—figure supplement 1: Quantitation missing on hts mutant @ 24h APF, need to compare with gogo and double mutants.

We have added the quantification of the *hts* mutant to Figure 5—figure supplement 1B. However, it is very technically demanding to put all of these components together in a single fly to create the double mutant.

e) Figure 5—figure supplement 1: No mention of panels E-G in the text.

We have mentioned these panels in the Discussion section.

Fig6:a) Panel A: Quantitation of dinr mutant needed.

We have added the quantification of the dinr mutant to Figure 6C.

b) Panel B: Authors should split the image to show the repo+ cells only. Because one cannot see the Repo+ red signal in the 2 cells (indicated by the top arrowheads).

We have split the image in Figure 6D.

c) Panels C-J: It would be helpful to know when these drivers come on. Could be differences in effects due to early vs. late expression in glia.

These driver promoters are ‘on’ from the larval stages (since it is controlled by shi[ts], these effects occur from APF0). Therefore, we do not anticipate that there are any differences in effects due to early vs late expression. This information was added to the Figure 6 legend.

d) hob RNAi quantitation should be included.

We have added the quantification of hobbit RNAi to Figure 6M.

Fig7:a) Panel A: show with 24B10 where the R7 layer is, so that one can see the position of the endings of the glial process that seem to be positioned below the R8 layer.

In the third instar larvae, R7 axons are well behind (above) the R8 axons spatially, so we believe that showing R8 is sufficient.

b) Panel B: quantitate co-localization with fmi and glial marker. This is very hard to interpret.

We have added the visual quantification of the localization of Fmi in glial cells (or R cells) as Figure 7B’.

c) Panels C-H: No quantitation shown. Difficult to see what the authors are trying to show here.

We have added this quantification as Figure 7E.

Also, the authors should show horizontal orientation for glial specific knockdown of fmi in mid-pupae or adult.

We have added the images of APF24% as Figure 7C’, D’.

d) Panel D: Columns are disorganized in glial specific knockdown of fmi. So, R8 positioning could be a result of the columnar disorganization rather than fmi instructing R8.

According to the Trush et al., (2019), R8 is one of the earliest axons that enters the medulla column. Only Mi1 is known to arrive earlier than R8, which makes it hard to imagine that R8 is already affected by the columnar structure which is shaped by R8 itself. It is more natural to interpret that glial Fmi guides R8 axons through homophilic interaction mediated by Fmi.

Reviewer #3:I am reviewing this paper with the view that it is probably not an option to perform experiments at this time, while still making sure that the authors conclusions are well supported by the data presented in the paper. This is why I do not suggest additional experiments..This is an interesting study by a group that are experts in studying how the transmembrane proteins Gogo and Fmi regulate targeting of the R8 axon in the fly visual system. The present work builds upon previous studies from the lab. Overall the data quality is excellent. The key observation here is that the phosphorylation status of Gogo seems to act as a switch and is required during distinct phases of R8 projection. Secretion from glial cells, and also Fmi expression in these cells, is required for proper R8 projection. These observations are interesting. However, the study suffers from significant draw backs that unfortunately prevent me from supporting it for publication at this stage.Firstly, this study presents significant overlap with a previous paper from the lab (Mann, 2012) that reported Gogo phosphorylation/dephosphorylation regulation by the insulin pathway, and showed that this is required for proper R8 projection in the fly visual system. While the present work looks at how these phenotypes develop through time, the idea that phosphorylation of Gogo might be regulated by Dilps and that it might change Gogo's function during R8 pathfinding is not new.

We agree with the reviewer that the current study has some overlap with the former publication, as this study is a continuation of the previous study; however, the current manuscript has made the following significant conceptual advances:

1) Gogo and Fmi protein localizations are now visualized.

2) As we have obtained a new marker for R8 during pupal stages, the true function of *gogo* in developmental stages is now known to be recognizing the column center (glia) in phase 1, and antagonizing filopodia extension in phase 2. These functions are specifically controlled by the phosphorylated species of Gogo.

3) New functionality for Fmi in column center glia has been discovered.

4) The source and ligand are now revealed as Dilp6 from glial cells.

In addition, the genetic evidence of *dilp6* from glial cells as the ligand is newly added to the current version of the manuscript. We hope altogether that the current form of the manuscript contains enough evidence to not only understand the function of Gogo and R8 axon pathfinding, but to convince the reviewers of the elucidated mechanisms of axon guidance and layer-specific targeting in the complex nervous system.

I also think that there is a need to better define and present the evidence for a switch in function for Gogo with respect to Fmi. Similarly, the authors should also make it clearer the extent to which Steps 1-3 are interdependent (or not). Isn't it possible that if step 1 fails, then 2 and 3 will also fail as a consequence? This is important because the authors tend to present these steps as relatively independent, but it is not made clear why they hold that view.

This is a valid point, and we have addressed this issue by generating a phase 2-specific knockdown of *gogo*, as explained above.

We still believe that phase 2 exists, and that it is required for normal R8 axon targeting, based on the following 4 lines of evidence:

1) Glial *fmi* loss-of-function mutation disrupted the horseshoe shape at phase 1, but not bundling at phase 2. We have added this data to Figure 7C’ and D’.

2) The nonphosphorylated GogoFFD rescue was not perfect, and there was still some axonal bundling, shown in Figure 5O.

3) We also added an experiment suggested below by the reviewer: the phase 2-specific *gogo* RNAi knockdown experiment using Gal80^ts^. As we anticipated, the *gogo* LOF phenotype, characterized by longer filopodia in more radial directions, was also observed in the phase 2-specific *gogo* knockdown, suggesting that the defects of Gogo LOF in phase 2 are independent of phase 1. We describe these results in Figure 2O-P''.

4) The DInR LOF mutant does not have defects in phase 1 (horseshoe shape), but exhibits a bundling phenotype in phase 2 (Figure 6A and Figure 6B, respectively). Therefore, there exists a signaling molecule that specifically regulates only phase 2, but not phase 1.

Point (1) above shows that in phase 1-specific disruption of Gogo-Fmi interaction, there is a disorganized array of R8 axons (Figure 7D’’), but not R8 bundling or invasion into the medulla layers. We think this is the phenotype of phase 1-specific disruption of *gogo* function, while point (3) demonstrates the phase 2-specific disruption of *gogo* function (Figure 2O-P’’). There, we see longer filopodia expanding in more radial directions, although the phenotype became milder because of residual Gal80 function and later onset of RNAi knockdown. The adult phenotype of *gogo* mutants may represent the addition of these phenotypes of phases 1 and 2. The disorganized array will enhance the R8 bundling and invasion when the filopodia become longer in random directions. However, from the experimental evidence, we show that the *gogo* functions in phase 1 and phase 2 are independent of each other inside the growth cone.

The work also feels very qualitative, especially Figure 1, and Figure 4. The authors conclusion would be greatly strengthened by providing quantifications next to the panels. I feel that given this is a relatively well-studied model system, the authors should try to come up with a better way to quantify the shape and location of the R8 termini. This would make it convincing and could even reveal new interesting features of the phenotype (and wild type).

This is a valid point, and we have addressed this issue by quantifying most of the phenotypes in Figure 1, Figure 4, and Figure 7, as was also suggested by other reviewers.